There are amendments to this paper

# Genetic variants of calcium and vitamin D metabolism in kidney stone disease

Sarah A. Howles [1,2,7]*, Akira Wiberg [3,7], Michelle Goldsworthy [1,2], Asha L. Bayliss[2], Anna K. Gluck [2], Michael Ng [3], Emily Grout[1], Chizu Tanikawa[4], Yoichiro Kamatani [5], Chikashi Terao [5], Atsushi Takahashi [5], Michiaki Kubo[5], Koichi Matsuda [6], Rajesh V. Thakker [2], Benjamin W. Turney[1] & Dominic Furniss [3]

Kidney stone disease (nephrolithiasis) is a major clinical and economic health burden with a heritability of ~45–60%. We present genome-wide association studies in British and Japanese populations and a trans-ethnic meta-analysis that include 12,123 cases and 417,378 controls, and identify 20 nephrolithiasis-associated loci, seven of which are previously unreported. A *CYP24A1* locus is predicted to affect vitamin D metabolism and five loci, *DGKD, DGKH, WDR72, GPIC1,* and *BCR*, are predicted to influence calcium-sensing receptor (CaSR) signaling. In a validation cohort of only nephrolithiasis patients, the *CYP24A1*-associated locus correlates with serum calcium concentration and a number of nephrolithiasis episodes while the *DGKD*-associated locus correlates with urinary calcium excretion. In vitro, DGKD knockdown impairs CaSR-signal transduction, an effect rectified with the calcimimetic cinacalcet. Our findings indicate that studies of genotype-guided precision-medicine approaches, including withholding vitamin D supplementation and targeting vitamin D activation or CaSR-signaling pathways in patients with recurrent kidney stones, are warranted.

[1] Nuffield Department of Surgical Sciences, University of Oxford, Oxford, UK. [2] Academic Endocrine Unit, Radcliffe Department of Medicine, University of Oxford, Oxford, UK. [3] Nuffield Department of Orthopaedics, Rheumatology and Musculoskeletal Sciences, University of Oxford, Oxford, UK. [4] Laboratory of Genome Technology, Human Genome Centre, University of Tokyo, Tokyo, Japan. [5] RIKEN Centre for Integrative Medical Sciences, Yokohama, Kanagawa, Japan. [6] Laboratory of Clinical Genome Sequencing, Department of Computational Biology and Medical Sciences, University of Tokyo, Tokyo, Japan. [7] These authors contributed equally: Sarah A. Howles, Akira Wiberg *email: sarah.howles@nds.ox.ac.uk

Kidney stones affect ~20% of men and ~10% of women by 70 years of age[1] and commonly cause debilitating pain. The prevalence of this disorder is increasing and the United States is predicted to spend over $5 billion per year by 2030 on its treatment[2]. Unfortunately, up to 50% of individuals will experience a second kidney stone episode within 10 years of their initial presentation[3] and recurrent stone disease is linked to renal function decline[4].

Twin studies have reported a heritability of > 45% and > 50% for stone disease and hypercalciuria, respectively[5,6] and a strong family history of urolithiasis, including a parent and a sibling, results in a standard incidence ratio (SIR) for stone formation of > 50 in contrast to a SIR of 1.29 in spouses[7]. Four genome-wide association studies of nephrolithiasis have been published identifying fifteen loci associated with disease[8–11]; however no trans-ethnic studies have been undertaken.

To increase understanding of the common genetic factors contributing to risk of nephrolithiasis, we present a genome-wide association study (GWAS) using the UK Biobank resource[12] and perform a subsequent meta-analysis with the summary statistics from the Biobank Japan nephrolithiasis genome-wide association study[11,13] to identify 20 loci associated with nephrolithiasis. One such locus is associated with *CYP24A1* and is predicted to affect vitamin D metabolism and five loci, *DGKD, DGKH, WDR72, GPIC1*, and *BCR*, are predicted to influence calcium-sensing receptor (CaSR) signaling. In a validation cohort of nephrolithiasis patients, we find that the *CYP24A1*-associated locus correlates with serum calcium concentration and number of kidney stone episodes and that the *DGKD*-associated locus correlates with urinary calcium excretion. Moreover, DGKD knockdown impairs CaSR-signal transduction in vitro, an effect that is rectified by the calcimimetic cinacalcet, thereby supporting the role of DGKD in CaSR signaling. Our findings suggest that further studies into the utility of genotyping to inform risk of incident kidney stone disease prior to vitamin D supplementation and to guide precision-medicine approaches, by targeting CaSR-signaling or vitamin D activation pathways in patients with recurrent kidney stones, are warranted.

## Results

**Association studies and in silico analyses.** Genome-wide association studies were undertaken in UK and Japanese populations. Stone forming individuals were identified from the UK Biobank using ICD-10 and OPCS codes and a single UK Biobank self-reported operation code for kidney stone surgery and excluded if recorded to have a disorder known to predispose to kidney stone disease (Supplementary Table 1 and 2). In contrast, Biobank Japan cases were diagnosed and enrolled by physicians; information regarding conditions known to predispose to kidney stones was unavailable. UK Biobank genome-wide association analysis was undertaken across 547,011 genotyped SNPs and ~8.4 million imputed SNPs with MAF ≥ 0.01 and Info Score ≥ 0.9, using a linear mixed non-infinitesimal model implemented in BOLT-LMM v2.3[14]. Biobank Japan GWAS was conducted using a logistic regression model incorporating age, sex, and the top 10 principal components as covariates. Subsequently, a trans-ethnic meta-analysis was performed using the summary statistics from the UK and Japanese GWAS data sets to integrate data from 12,123 stone formers and 417,378 controls.

Twenty genetic loci associating with nephrolithiasis were identified, 10 of which were initially identified from the UK Biobank discovery cohort (Supplementary Table 3) and another 10 from the subsequent trans-ethnic meta-analysis with Japanese GWAS summary statistics (Table 1, Fig. 1, and Supplementary Fig. 1)[11]. Seven of the identified loci have not previously been

reported to associate with kidney stone disease at GWAS (Table 1)[8–11,15,16]. The allelic effects were concordant across both studies at all 20 loci, with minimal evidence of heterogeneity between the two GWAS at the majority of loci, with a Q-statistic p-value > 0.05 at 17 out of 20 loci (Supplementary Table 4), suggesting that the genetic architecture of kidney stone disease is very similar between populations of European and East Asian ancestry. For the seven previously unreported loci, we found that the effects are directionally concordant in both datasets and of the same magnitude, with the exception of rs13054904 (the *BCR* locus). This locus therefore highlights ethnic differences in the predisposition to renal stone disease.

Out of 849 SNPs with a meta-analysis significant p-value of $p < 5.0 \times 10^{-8}$, many demonstrated evidence of potential functionality: 33 SNPs had a combined annotation-dependent depletion (CADD) score > 12.37, the threshold suggested for deleterious SNPs[17]. A further 54 SNPs had a RegulomeDB score of 2b or higher, which is likely to affect protein binding[18] (Supplementary Data 1).

Fifty-four candidate genes were identified via in silico analysis of these 20 loci based on FUMA positional mapping, functional annotation, and biological plausibility[19]. MAGMA gene-property analysis implemented in FUMA revealed a striking overexpression of these genes in the kidney cortex; the GENE2FUNC tool demonstrated enrichment for gene ontologies associated with transmembrane ion transport, renal function, and calcium homeostasis, including response to vitamin D (Supplementary Figs. 2–3 and Supplementary Table 5)[20].

**CYP24A1 associated locus.** We observed a genome-wide significant signal at a locus (rs17216707) that is ~38 kb upstream of *CYP24A1*, a gene encoding cytochrome P450 family 24 subfamily A member 1 (CYP24A1), an enzyme that metabolizes active 1,25-dihydroxyvitamin D to inactive 24,25-dihydroxyvitamin D. Loss-of-function mutations in CYP24A1 cause autosomal recessive infantile hypercalcaemia type 1 (OMIM 126065)[21]. We postulated that the *CYP24A1* increased-risk allele associates with decreased CYP24A1 activity, leading to perturbations of calcium homeostasis and mimicking an attenuated form of infantile hypercalcaemia type 1. Therefore, associations of rs17216707 with serum calcium, phosphate, parathyroid hormone (PTH), and 25-hydroxyvitamin D concentrations, and urinary calcium excretion, and number of kidney stone episodes were sought in a validation cohort of 440 kidney stone formers attending the Oxford University Hospitals NHS Foundation Trust for treatment of kidney stones. Only stone forming patients were recruited, 15% of individuals were taking medications, including steroids and diuretics, that may affect risk of kidney stone formation. 1,25-hydroxyvitamin D levels were unavailable. Reference ranges for 24-hour urinary calcium excretion differ for men and women[22], thus associations with urinary calcium excretion were examined separately.

Individuals homozygous for the *CYP24A1* increased-risk allele rs17216707 (T) had a significantly increased mean serum calcium concentration when compared to heterozygotes (mean serum calcium 2.36 mmol/l (TT) *vs.* 2.32 mmol/l (TC); trend test p-value for additive effect = 0.023) (Table 2). Analysis of the allelic frequencies of this SNP between cases and controls within the UK Biobank cohort was not able to provide definitive evidence for the recessive model over other genetic models (Supplementary Table 6). rs17216707 (T) homozygotes had more kidney stone episodes than heterozygotes (mean number of stone episodes 4.0 (TT) *vs.* 2.4 (TC), Mann-Whitney U-test $p = 0.0003$) and there was a significant correlation across genotypes (TT *vs.* TC *vs.* CC) with number of stone episodes (Kruskal–Wallis $p = 0.0024$)

**Table 1 SNPs significantly associated with kidney stone disease at trans-ethnic meta-analysis**

| Ch[a] | SNP | Position[b] | Ann[c] | EA[d] | NEA[e] | | EAF[f] | INFO[g] | OR[h] | p[i] | Candidate gene |
|---|---|---|---|---|---|---|---|---|---|---|---|
| 1 | rs10917002 | 21836340 | I | T | C | UK[j] | 0.11 | 0.997 | 1.18 (1.12–1.25) | **$3.60 \times 10^{-9}$** | ALPL[†] |
|  |  |  |  |  |  | BJ[k] | 0.38 | 0.998 | 1.09 (1.04–1.15) | $5.83 \times 10^{-5}$ |  |
|  |  |  |  |  |  | MA[l] |  |  | 1.13 (1.09–1.17) | **$3.45 \times 10^{-11}$** |  |
| 2 | rs780093 | 27742603 | I | T | C | UK[j] | 0.38 | — | 1.08 (1.04–1.12) | $3.60 \times 10^{-5}$ | GCKR |
|  |  |  |  |  |  | BJ[k] | 0.56 | 1 | 1.14 (1.09–1.18) | **$1.10 \times 10^{-8}$** |  |
|  |  |  |  |  |  | MA[l] |  |  | 1.10 (1.08–1.13) | **$1.31 \times 10^{-13}$** |  |
| 2 | rs13003198* | 234257105 | IG | T | C | UK[j] | 0.39 | 0.997 | 1.10 (1.06–1.14) | $6.50 \times 10^{-8}$ | DGKD |
|  |  |  |  |  |  | BJ[k] | 0.25 | 0.98 | 1.12 (1.06–1.18) | $1.09 \times 10^{-5}$ |  |
|  |  |  |  |  |  | MA[l] |  |  | 1.11 (1.07–1.14) | **$3.89 \times 10^{-11}$** |  |
| 4 | rs1481012* | 89039082 | I | G | A | UK[j] | 0.11 | 0.994 | 1.12 (1.06–1.18) | $4.30 \times 10^{-5}$ | ABCG2 |
|  |  |  |  |  |  | BJ[k] | 0.30 | 0.994 | 1.11 (1.05–1.17) | $1.50 \times 10^{-5}$ |  |
|  |  |  |  |  |  | MA[l] |  |  | 1.11 (1.07–1.16) | **$2.79 \times 10^{-8}$** |  |
| 5 | rs56235845 | 176798040 | S | G | T | UK[j] | 0.33 | 0.986 | 1.16 (1.12–1.20) | **$9.10 \times 10^{-15}$** | SLC34A1 |
|  |  |  |  |  |  | BJ[k] | 0.31 | 0.87 | 1.18 (1.12–1.25) | **$1.88 \times 10^{-11}$** |  |
|  |  |  |  |  |  | MA[l] |  |  | 1.16 (1.13–1.20) | **$2.64 \times 10^{-21}$** |  |
| 6 | rs1155347 | 39146230 | IG | C | T | UK[j] | 0.22 | 0.975 | 1.12 (1.07–1.17) | $2.60 \times 10^{-7}$ | KCNK5 |
|  |  |  |  |  |  | BJ[k] | 0.16 | 0.925 | 1.16 (1.08–1.24) | $1.33 \times 10^{-6}$ |  |
|  |  |  |  |  |  | MA[l] |  |  | 1.13 (1.09–1.17) | **$8.54 \times 10^{-11}$** |  |
| 6 | rs77648599* | 160624115 | I | G | T | UK[j] | 0.03 | 0.992 | 1.33 (1.21–1.47) | **$5.50 \times 10^{-9}$** | SLC22A2 |
|  |  |  |  |  |  | BJ[k] | 0.04 | 0.739 | 1.22 (1.06–1.44) | $1.89 \times 10^{-3}$ |  |
|  |  |  |  |  |  | MA[l] |  |  | 1.30 (1.20–1.42) | **$5.39 \times 10^{-10}$** |  |
| 7 | rs12539707* | 27626165 | I | T | C | UK[j] | 0.30 | 0.999 | 1.13 (1.08–1.17) | **$6.30 \times 10^{-10}$** | HIBADH |
|  |  |  |  |  |  | BJ[k] | 0.09 | 0.789 | 1.10 (1.01–1.21) | 0.0268 |  |
|  |  |  |  |  |  | MA[l] |  |  | 1.12 (1.08–1.16) | **$1.09 \times 10^{-10}$** |  |
| 7 | rs12666466 | 30916430 | I | G | C | UK[j] | 0.03 | 0.994 | 1.22 (1.11–1.34) | $5.00 \times 10^{-5}$ | AQP1 |
|  |  |  |  |  |  | BJ[k] | 0.12 | 0.989 | 1.17 (1.08–1.26) | $2.80 \times 10^{-6}$ |  |
|  |  |  |  |  |  | MA[l] |  |  | 1.19 (1.12–1.26) | **$3.26 \times 10^{-8}$** |  |
| 11 | rs4529910 | 111243102 | I | T | G | UK[j] | 0.27 | 0.998 | 1.07 (1.02–1.11) | $1.40 \times 10^{-3}$ | POU2AF |
|  |  |  |  |  |  | BJ[k] | 0.59 | 0.999 | 1.12 (1.08–1.16) | $3.94 \times 10^{-7}$ |  |
|  |  |  |  |  |  | MA[l] |  |  | 1.09 (1.06–1.12) | **$4.25 \times 10^{-10}$** |  |
| 13 | rs1037271 | 42779410 | I | C | T | UK[j] | 0.39 | 0.995 | 1.20 (1.15–1.24) | **$2.50 \times 10^{-8}$** | DGKH[†] |
|  |  |  |  |  |  | BJ[k] | 0.55 | 0.936 | 1.15 (1.12–1.18) | **$7.49 \times 10^{-15}$** |  |
|  |  |  |  |  |  | MA[l] |  |  | 1.17 (1.15–1.20) | **$1.29 \times 10^{-24}$** |  |
| 15 | rs578595 | 53997089 | I | C | A | UK[j] | 0.46 | 0.996 | 1.09 (1.05–1.13) | $2.50 \times 10^{-6}$ | WDR72 |
|  |  |  |  |  |  | BJ[k] | 0.69 | 0.996 | 1.11 (1.05–1.15) | $2.25 \times 10^{-5}$ |  |
|  |  |  |  |  |  | MA[l] |  |  | 1.09 (1.07–1.12) | **$6.26 \times 10^{-11}$** |  |
| 16 | rs77924615 | 20392332 | I | A | G | UK[j] | 0.20 | 0.980 | 1.13 (1.08–1.18) | **$1.80 \times 10^{-8}$** | UMOD |
|  |  |  |  |  |  | BJ[k] | 0.22 | 0.984 | 1.17 (1.10–1.24) | **$2.80 \times 10^{-9}$** |  |
|  |  |  |  |  |  | MA[l] |  |  | 1.14 (1.10–1.19) | **$1.14 \times 10^{-13}$** |  |
| 16 | rs889299* | 23381914 | I | G | A | UK[j] | 0.76 | 1 | 1.10 (1.05–1.14) | $8.20 \times 10^{-6}$ | SCNN1B |
|  |  |  |  |  |  | BJ[k] | 0.66 | 0.895 | 1.09 (1.04–1.14) | $9.39 \times 10^{-4}$ |  |
|  |  |  |  |  |  | MA[l] |  |  | 1.09 (1.06–1.13) | **$1.55 \times 10^{-8}$** |  |
| 17 | rs1010269 | 59448945 | I | G | A | UK[j] | 0.83 | 0.981 | 1.08 (1.03–1.14) | $7.10 \times 10^{-4}$ | BCAS[†] |
|  |  |  |  |  |  | BJ[k] | 0.56 | 0.87 | 1.17 (1.12–1.22) | **$4.82 \times 10^{-11}$** |  |
|  |  |  |  |  |  | MA[l] |  |  | 1.13 (1.10–1.17) | **$3.71 \times 10^{-15}$** |  |
| 17 | rs4793434* | 70352537 | I | G | C | UK[j] | 0.50 | 0.993 | 1.09 (1.05–1.13) | $1.50 \times 10^{-6}$ | SOX9 |
|  |  |  |  |  |  | BJ[k] | 0.32 | 0.983 | 1.09 (1.04–1.15) | $2.04 \times 10^{-4}$ |  |
|  |  |  |  |  |  | MA[l] |  |  | 1.09 (1.06–1.12) | **$4.52 \times 10^{-9}$** |  |
| 19 | rs3760702 | 14588237 | IG | A | G | UK[j] | 0.33 | 0.994 | 1.08 (1.05–1.13) | $1.40 \times 10^{-5}$ | GIPC1 |
|  |  |  |  |  |  | BJ[k] | 0.25 | 0.971 | 1.14 (1.08–1.20) | $3.78 \times 10^{-7}$ |  |
|  |  |  |  |  |  | MA[l] |  |  | 1.09 (1.07–1.13) | **$1.98 \times 10^{-9}$** |  |
| 20 | rs17216707 | 52732362 | IG | T | C | UK[j] | 0.81 | 0.961 | 1.17 (1.12–1.22) | **$9.90 \times 10^{-12}$** | CYP24A1 |
|  |  |  |  |  |  | BJ[k] | 0.92 | 0.766 | 1.24 (1.15–1.34) | $5.90 \times 10^{-6}$ |  |
|  |  |  |  |  |  | MA[l] |  |  | 1.19 (1.14–1.23) | **$7.82 \times 10^{-18}$** |  |
| 21 | rs12626330 | 37835982 | I | G | C | UK[j] | 0.49 | 0.980 | 1.16 (1.12–1.20) | **$5.80 \times 10^{-17}$** | CLDN14 |
|  |  |  |  |  |  | BJ[k] | 0.39 | 0.981 | 1.12 (1.07–1.18) | $2.77 \times 10^{-7}$ |  |
|  |  |  |  |  |  | MA[l] |  |  | 1.15 (1.12–1.18) | **$7.24 \times 10^{-21}$** |  |
| 22 | rs13054904* | 23410918 | I | A | T | UK[j] | 0.26 | 0.999 | 1.15 (1.11–1.20) | **$3.30 \times 10^{-12}$** | BCR |
|  |  |  |  |  |  | BJ[k] | 0.02 | 0.967 | 1.05 (0.91–1.26) | 0.505 |  |
|  |  |  |  |  |  | MA[l] |  |  | 1.14 (1.10–1.19) | **$4.49 \times 10^{-12}$** |  |

[a]Chromosome. [b]Based on NCBI Genome Build 37 (hg19). [c]Annotation. I denotes an intronic position, IG an intergenic position, and S a splice site position. [d]The effect allele. [e]The alternate (non-effect) allele. [f]The effect allele frequency in the study population. [g]The imputation quality score. [h]Odds ratio (95% confidence intervals). OR >1 indicative of increased risk with effect allele. [i]p values less than the genome-wide significance threshold of $5.0 \times 10^{-8}$ are shown in bold italics. [j]UK Biobank, cohort 6536 cases and 388,508 controls. [k]Biobank Japan, cohort 5587 cases and 28,870 controls. [l]Trans-ethnic meta-analysis. *Loci not previously reported to associate with kidney stone disease in GWAS. [†]These three variants showed nominally significant heterogeneity in effects between the two populations (Q value p < 0.05).

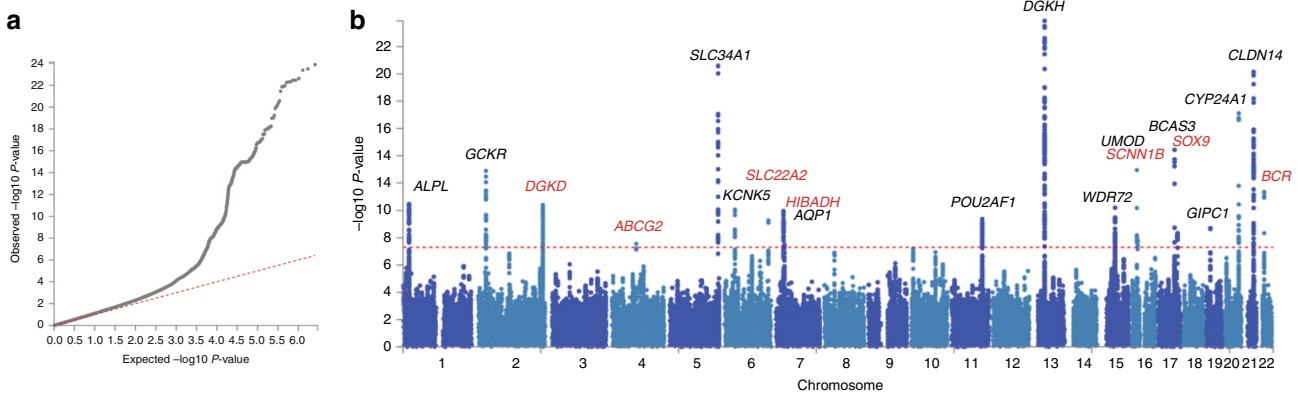

**Fig. 1** Results of trans-ethnic genome-wide association study in kidney stone disease. A trans-ethnic meta-analysis of kidney stone disease was performed for 12,123 individuals with kidney stone disease and 417,378 controls from the UK Biobank and BioBank Japan. **a** is a quantile-quantile plot of observed *vs.* expected p-values. The $\lambda_{GC}$ demonstrated some inflation (1.0957), but the LD score regression (LDSC) intercept of 0.9997, with an attenuation ratio of 0.0075 indicated that the inflation was largely due to polygenicity and the large sample size. **b** is a Manhattan plot showing the genome-wide p values (-log10) plotted against their respective positions on each of the autosomes. The horizontal red line shows the genome-wide significance threshold of $5.0 \times 10^{-8}$. Loci have been labeled with the primary candidate gene at each locus, as shown in Table 1. Previously unreported GWAS-discovered kidney stone loci are highlighted in red

(Table 2). No correlation was found between rs17216707 genotype and serum phosphate, PTH, 25-hydroxyvitamin D concentration or urinary calcium excretion. Intake of medications including steroids and diuretics was comparable across genotypes (Supplementary Fig. 4A).

**Calcium-sensing receptor signaling pathway associated loci.** Five of the identified loci are linked to genes that are predicted to influence CaSR signaling (*DGKD, DGKH, WDR72, GIPC1,* and *BCR*) (Fig. 2). rs13003198 is ~6 kb upstream of *DGKD*, encoding diacylglycerol kinase delta (DGKD); rs1037271 is an intronic variant in *DGKH*, encoding diacylglycerol kinase eta (DGKH). DGKD and DGKH phosphorylate diacylglcerol, a component of the intracellular CaSR-signaling pathway inducing CaSR-mediated membrane ruffling and activating protein kinase C signaling cascades including MAPK and intracellular calcium release[23,24] (Fig. 2). rs578595 is an intronic variant in *WDR72* encoding WD repeat domain 72 (WDR72) and rs3760702 is ~300 bp upstream of *GIPC1* that encodes Regulator of G-protein signaling 19 Interacting Protein 1 (GIPC1). Both WDR72 and GIPC1 are thought to play a role in clathrin-mediated endocytosis, a process central to sustained intracellular CaSR signaling[24–27] (Fig. 2). rs13054904 is ~110 kb upstream of *BCR*, encoding a RAC1 (Rac Family Small GTPase 1) GTPase-activating protein known as Breakpoint Cluster Region (BCR)[28]. RAC1 activation is induced by CaSR ligand binding and mediates CaSR-induced membrane ruffling[23] (Fig. 2). Of the 20 genome-wide significant loci discovered in the meta-analysis of British and Japanese populations, this was the only locus that did not reach nominal significance in one of the two populations (British: $p = 3.30 \times 10^{-12}$; Japanese: $p = 0.505$), suggesting that this locus predisposes to renal stones in European but not East Asian populations. In addition, a previously reported association between the CaSR-associated intronic SNP rs7627468 and nephrolithiasis was confirmed ($p = 3.5 \times 10^{-5}$)[9].

Gain-of-function mutations in components of the CaSR-signaling pathway result in autosomal dominant hypocalcaemia (ADH, OMIM 601198, 615361), which is associated with hypercalciuria in ~10% of individuals[29,30]. ADH-associated mutations result in a gain-of-function in CaSR-intracellular signaling in vitro via pathways including intracellular calcium ions and MAPK[29,31,32]. We hypothesized that the nephrolithiasis-

associated loci linked to CaSR-signaling associate with enhanced CaSR signal transduction resulting in a biochemical phenotype mimicking an attenuated form of ADH. *DGKD* and *DGKH* were selected for further analysis based on their potential to influence both membrane ruffling and protein kinase C CaSR-signaling pathways (Fig. 2).

The *DGKD* increased-risk allele rs838717 (G) (top *DGKD*-associated SNP in the UK Biobank GWAS, Supplementary Table 3, linkage disequilibrium with rs13003198 $r^2 = 0.53$) associated with increased 24-h urinary calcium excretion in male stone formers (mean 24-h urinary calcium excretion 7.27 mmol (GG) *vs.* 4.54 mmol (AA); trend test *p*-value for additive effect = 0.017) (Table 2), consistent with enhanced CaSR-signal transduction. No association of the *DGKD* increased-risk allele rs838717 (G) with urinary calcium excretion was identified in female stone formers. No correlations were observed between genotype and serum calcium, phosphate, PTH, 25-hydroxyvitamin D concentrations or number of stone episodes.

The *DGKH* increased-risk allele rs1170174 (A) (top *DGKH*-associated SNP in the UK Biobank GWAS, Supplementary Table 3, linkage disequilibrium with rs13003198 $r^2 = 0.3$) did not associate with biochemical phenotype or stone recurrence. However, the sample size of homozygotes was small (AA, $n = 6$) and prior to Bonferroni correction, a suggestive association was detected with urinary calcium excretion in male stone formers (mean 24-h urinary calcium excretion 8.14 mmol (AA) *vs.* 5.09 mmol (GG), student's *t* test $p = 0.0503$) (Supplementary Table 7). Prescription of medications, including steroids and diuretics, was comparable across *DGKD* and *DGKH* genotypes (Supplementary Fig. 4B, 4C).

To investigate the role of DGKD in CaSR-signaling via the MAPK pathway, HEK-CaSR-SRE and HEK-CaSR cells were treated with scrambled or *DGKD* targeted siRNA and intracellular MAPK responses to alterations in extracellular calcium concentration assessed via SRE and ERK-phosphorylation (pERK) assays, respectively. Treatment with *DGKD* targeted siRNA resulted in a reduction in DGKD expression when compared to cells treated with scrambled siRNA without alteration in CaSR expression (Fig. 3a–c and Supplementary Fig. 5A, B). SRE and pERK responses were significantly decreased in cells with reduced DGKD expression (DGKD-KD) when compared to cells with baseline DGKD expression (WT) (SRE maximal response DGKD-KD = 5.28 fold change, 95% confidence interval (CI) = 4.77–5.79

**Table 2 Genotype-phenotype correlations in cohort of kidney stone formers**

| Variable | Normal range§ | CYP24A1 (rs17216707) TT | TC | CC | DGKD (rs838717) AA | AG | GG |
|---|---|---|---|---|---|---|---|
| **Serum** | | | | | | | |
| Calcium (mmol/l) | 2.10–2.50 | **2.36 ± 0.01\* (260)** | **2.32 ± 0.01 (109)** | 2.34 ± 0.02 (15) | 2.34 ± 0.01 (107) | 2.35 ± 0.01 (182) | 2.36 ± 0.01 (95) |
| Phosphate (mmol/l) | 0.7–1.40 | 1.02 ± 0.13 (274) | 1.02 ± 0.02 (111) | 0.99 ± 0.05 (14) | 1.03 ± 0.02 (114) | 1.01 ± 0.02 (193) | 1.02 ± 0.02 (92) |
| Parathyroid hormone (pmol/l) | 1.3–7.6 | 5.06 ± 0.18 (271) | 5.59 ± 0.32 (107) | 5.27 ± 0.64 (14) | 5.34 ± 0.32 (108) | 5.38 ± 0.23 (189) | 4.72 ± 0.23 (95) |
| 25-hydroxy vitamin D (nmol/l) | >50 | 54.8 ± 1.84 (227) | 50.6 ± 2.60 (89) | 55.7 ± 6.92 (10) | 56.1 ± 3.13 (88) | 52.6 ± 2.04 (157) | 53.2 ± 2.90 (81) |
| **Urine** | | | | | | | |
| Male patients 24 hr calcium excretion (mmol) | <7.5 | 6.11 ± 0.47 (77) | 4.82 ± 0.55 (33) | 3.101.27 (3) | **4.54 ± 0.45\* (33)** | 5.45 ± 0.48 (57) | **7.27 ± 0.91 (25)** |
| Female patients 24 hr calcium excretion (mmol) | <6.2 | 4.85 ± 0.53 (31) | 5.06 ± 0.56 (15) | 4.27 ± 1.69 (2) | 5.83 ± 1.34 (9) | 4.92 ± 0.46 (27) | 4.12 ± 0.56 (12) |
| Number stone episodes | – | **4.0 ± 0.4\* (287)** | 2.4 ± 0.2 (119) | 2.4 ± 0.4 (14) | 3.5 ± 0.41 (114) | 3.8 ± 0.43 (208) | 2.8 ± 0.26 (98) |

A total of 440 patients were recruited, numbers of stone forming patients included in analysis are shown in parentheses. Serum calcium values are albumin-adjusted. All values are expressed as mean ± SEM. Student's t-tests were used for comparisons between groups of parametric data, Mann–Whitney U-tests were used for comparison of non-parametric data (number of stone episodes). ANOVA tests were used for comparisons of multiple sets of parametric data, no significance was reached. Kruskal–Wallis tests were used for comparisons of multiple sets of non-parametric data (number of stone episodes), significance at $p = 0.0024$ was reached for CYP24A1 locus correlations. Trend tests were performed for additive effects of rs17216707 on serum calcium ($p = 0.023$) and rs838717 on urinary calcium excretion ($p = 0.017$). Associations with biochemical phenotypes and number of stone episodes were sought at three loci rs17216707 (CYP24A1), rs838717 (DGKD) and rs1170174 (DGKH), no significant associations were detected, Supplementary Table 7). \*Denotes significance on comparison to bold cohort within group at Bonferroni corrected threshold of $p < 0.05/7 = 0.007$. §Normal ranges are from Nesbit et al.[29] and Curhan et al.[22]

*vs.* WT = 7.20 fold change, 95% CI = 6.46–7.93, F test $p = 0.0065$; pERK maximal response DGKD-KD = 24.77, 95% CI = 22.16–27.38 *vs.* WT = 39.46 fold change, 95% CI = 34.07–44.84, F test $p = 0.0056$). Cinacalcet rectified this loss-of-function in SRE-reporter assays (DGKD-KD + 5 nM cinacalcet, maximal response = 7.62 fold change, 95% CI = 5.98–9.27) (Fig. 3d–e and Supplementary Fig. 5C).

Furthermore, to investigate the role of DGKD in CaSR-signaling via intracellular calcium mobilization, HEK-CaSR-NFAT and HEK-CaSR cells were treated with scrambled or *DGKD* targeted siRNA and intracellular NFAT and intracellular calcium responses to alterations in extracellular calcium concentration assessed via NFAT-reporter and Fluo-4 calcium assays, respectively. Intracellular calcium responses were unaffected by a reduction in DGKD expression (NFAT maximal response DGKD-KD = 2.91 fold change, 95% CI = 2.69–3.12 *vs.* WT = 2.85 fold change, 95% CI = 2.62–3.08, F test $p = 0.73$; Fluo-4 maximal response DGKD-KD = 96.16, 95% CI = 89.83–102.5 *vs.* WT = 96.78 fold change, 95% CI = 92.76–100.8, F test $p = 0.83$) (Supplementary Fig. 5D, E).

## Discussion

Our study, which represents the largest kidney stone GWAS to date and integrates data from 12,123 stone formers and 417,378 controls from British and Japanese ancestries, identifies 20 genetic loci associated with nephrolithiasis, seven of which have not previously been reported to associate with kidney stone disease at GWAS. The genes implicated by our GWAS are disproportionately expressed in the renal cortex, with enrichment for biological pathways and gene ontologies involving solute transport, renal physiology and calcium homeostasis.

Our findings highlight the role of vitamin D catabolism in kidney stone formation. Thus, we identify a locus ~38 kb upstream of *CYP24A1* (rs17216707) and demonstrate that genotype at this locus is associated with serum calcium concentration and stone recurrence episodes in a cohort of kidney stone patients. These findings support our hypothesis that the rs17216707 increased-risk allele is associated with a relative hypercalcaemia and reduced activity of the 24-hydroxylase enzyme. Patients with loss-of-function CYP24A1 mutations have been successfully treated with inhibitors of vitamin D synthesis including fluconazole, similar therapies may be useful in rs17216707 (TT) recurrent kidney stone formers[33]. Furthermore, vitamin D supplementation in patients with biallelic loss-of-function mutations in CYP24A1 cause nephrocalcinosis[21]. We predict that rs17216707 (TT) individuals may similarly display an increased sensitivity to vitamin D. The National Institute of Clinical Excellence (NICE) recommends that all adults living in the UK should take daily vitamin D supplementation. However, our findings suggest that supplementation may put individuals homozygous for the *CYP24A1* increased-risk allele at risk of kidney stones; studies are required to investigate this hypothesis further.

Five of the loci identified at GWAS are linked to genes that are predicted to influence CaSR signaling: *DGKD, DGKH, WDR72, GIPC1,* and *BCR*. The CaSR is a G-protein coupled receptor that is highly expressed in the parathyroid and kidneys and has a central role in calcium homeostasis, increasing renal calcium reabsorption and stimulating PTH release to enhance bone resorption, urinary calcium reabsorption, and renal synthesis of 1,25-dihydroxyvitamin D[34].

The *DGKD* increased-risk allele associates with urinary calcium excretion in male stone formers but not in female stone formers. This is probably due to a lack of power as a result of the small sample size of female stone formers, however it is interesting to note that heritability of stone disease is lower in women than in men[5].

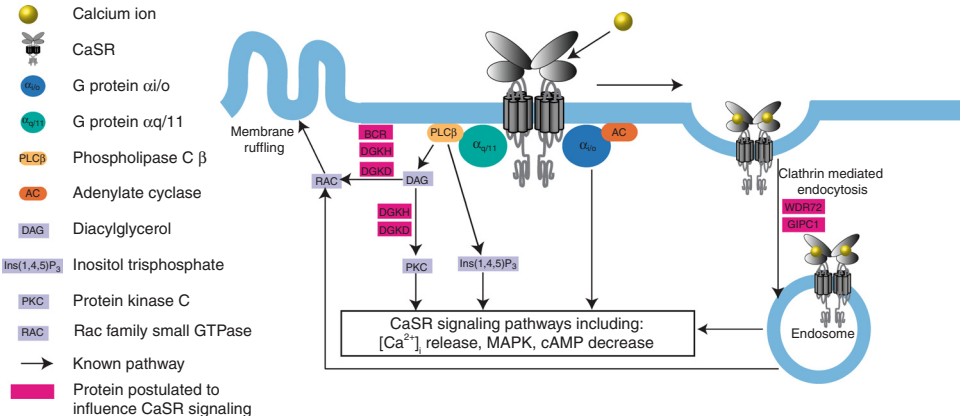

**Fig. 2** Schematic model for CaSR signaling. Ligand binding of calcium ions (yellow) by the G protein coupled receptor CaSR (gray) results in G protein-dependent stimulation via $G\alpha_{q/11}$ (turquoise) or $G\alpha_{i/o}$ (blue) causing stimulation of intracellular signaling pathways including intracellular calcium ($[Ca^{2+}]_i$) release, MAPK stimulation or cAMP reduction. $G\alpha_{q/11}$ signals via inositol 1,4,5-trisphosphate (IP3) and diacylglycerol (DAG). DAG leads to protein kinase C (PKC) stimulation along with RAC activation, which results in membrane ruffling. Following calcium ion binding the CaSR is internalized via clathrin-mediated endocytosis where signaling continues via the endosome. Proteins postulated to influence CaSR-signaling and their potential sites of action are shown in magenta

Furthermore, no correlations were identified with serum calcium concentrations despite this being the prominent phenotype in ADH patients. Reduced expression of DGKD results in decreased CaSR-mediated intracellular signaling via the MAPK pathway in vitro, but not via intracellular calcium mobilization. These findings provide evidence that DGKD influences CaSR-mediated signal transduction and suggest that the *DGKD* increased-risk allele may associate with a relative increase in DGKD expression thereby enhancing CaSR-mediated signaling via the MAPK pathway whilst leaving signaling via the intracellular calcium pathway unaffected. This biased signaling may provide an explanation for the observed correlation of the *DGKD* increased-risk allele rs838717 (G) with increased urinary calcium excretion but not serum calcium concentration (Table 2). Calcilytics, including NPS-2143 and ronacaleret, rectify enhanced CaSR-mediated signaling in vitro and biochemical phenotypes in mouse models of ADH[32,35,36]. We predict that the development of biased calcilytics may provide a novel, targeted therapeutic approach to reduce urinary calcium excretion in recurrent stone formers carrying CaSR-associated increased-risk alleles.

In conclusion, this study identifies 20 loci linked to kidney stone formation, seven of which are have not previously been reported to associate with kidney stone disease, and reveals the importance of vitamin D metabolism pathways and enhanced CaSR-signaling in the pathogenesis of nephrolithiasis. Our findings suggest that there may be a role for genetic testing to identify individuals in whom vitamin D supplementation should be used with caution and to facilitate a precision-medicine approach for the treatment of recurrent kidney stone disease, whereby targeting of the CaSR-signaling or vitamin D metabolism pathways may be beneficial in the treatment of a subset of patients with nephrolithiasis; further studies are required.

## Methods

**Study participants**. The UK Biobank resource was utilized for the UK-based GWAS. The UK Biobank is a prospective cohort study of ~500,000 individuals from the UK, aged between 40 and 69, who have had whole-genome genotyping undertaken, and have allowed linkage of these data with their medical records[12,37]. ICD-10 and OPCS codes and a single UK Biobank self-reported operation code for kidney stone surgery were used to identify individuals with a history of nephrolithiasis (Supplementary Table 1). Individuals were excluded if they were recorded to have a disorder of calcium homeostasis, malabsorption, or other condition known to predispose to kidney stone disease (Supplementary Table 2). Following quality control (QC) 6,536 UK Biobank participants were identified as cases and 388,508 individuals as controls (Supplementary Table 8).

In Japanese patients a diagnosis of nephrolithiasis was confirmed by enrolling physicians; patients with bladder stones were excluded. Information regarding conditions known to predispose to kidney stones was unavailable. DNA samples of 5,587 nephrolithiasis patients were obtained from BioBank Japan[13]. Controls (28,870 individuals) were identified from four population-based cohorts, including the JPHC (Japan Public Health Center)-based prospective study[38], the J-MICC (Japan Multi-Institutional Collaborative Cohort) study[39], IMM (Iwate Tohoku Medical Megabank Organization) and ToMMo (Tohoku Medical Megabank Organization)[40], (Supplementary Table 9).

Four hundred and forty patients attending the Oxford University Hospitals NHS Foundation Trust for treatment of kidney stones were enrolled into the Oxford University Hospitals NHS Foundation Trust Biobank of Kidney Stone Formers, following informed consent. These patients made up the validation cohort. Clinical data including urological history, past medical history, medication intake, and family history was recorded along with details of serum and urinary biochemistry. Whole blood and urine samples were stored at -80 °C.

**Ethical approval**. UK Biobank has approval from the North West Multi-Centre Research Ethics Committee (11/NW/0382), and this study (Epidemiology of Kidney Stone Disease) has UK Biobank study ID 885. Ethical committees at the Institute of Medical Science, The University of Tokyo and RIKEN approved the project (study IDs 29–74-A0215 and 17–17–16(3)). Collection of clinical data and biological samples from kidney stone patients attending the Oxford University Hospitals NHS Foundation Trust was approved by the University of Oxford under the Oxford Radcliffe Biobank research tissue bank ethics (09/H0606/5 + 5). All patients provided written informed consent.

**Genotyping**. The UK Biobank contains genotypes of 488,377 participants who were genotyped on two very similar genotyping arrays: UK BiLEVE Axiom Array (807,411 markers; 49,950 participants), and UK Biobank Axiom Array (825,927 markers; 438,427 participants). The two arrays are very similar, sharing ~95% of marker content. Genotypes were called from the array intensity data, in 106 batches of approximately 4700 samples each using a custom genotype-calling pipeline[41].

Japanese samples were genotyped with Illumina HumanOmniExpressExome BeadChip or a combination of the Illumina HumanOmniExpress and HumanExome BeadChips[42,43].

For genotyping of the validation cohort, DNA was extracted from whole blood samples using the Maxwell 16 Tissue DNA purification kit (Promega). SNP genotyping was undertaken using TaqMan SNP Genotyping Assays (ThermoFisher, Supplementary Table 11) and Type-it® Fast SNP Probe PCR Kit (Qiagen).

**Quality Control**. Quality Control (QC) of the UK Biobank data was performed using PLINK[44] v1.9 and R v3.3.1. All SNPs with a call rate < 90% were removed, accounting for the two different genotyping platforms used to genotype the individuals. Sample-level QC was undertaken and individuals excluded with one or more of the following: (1) call rate < 98%, (2) discrepancy between genetically inferred sex (Data Field 22001) and self-reported sex (Data Field 31), or individuals with sex chromosome aneuploidy (Data Field 22019), (3) heterozygosity >

Legend (Fig. 2):
- Calcium ion
- CaSR
- G protein αi/o
- G protein αq/11
- PLCβ  Phospholipase C β
- AC  Adenylate cyclase
- DAG  Diacylglycerol
- Ins(1,4,5)P3  Inositol trisphosphate
- PKC  Protein kinase C
- RAC  Rac family small GTPase
- → Known pathway
- Protein postulated to influence CaSR signaling

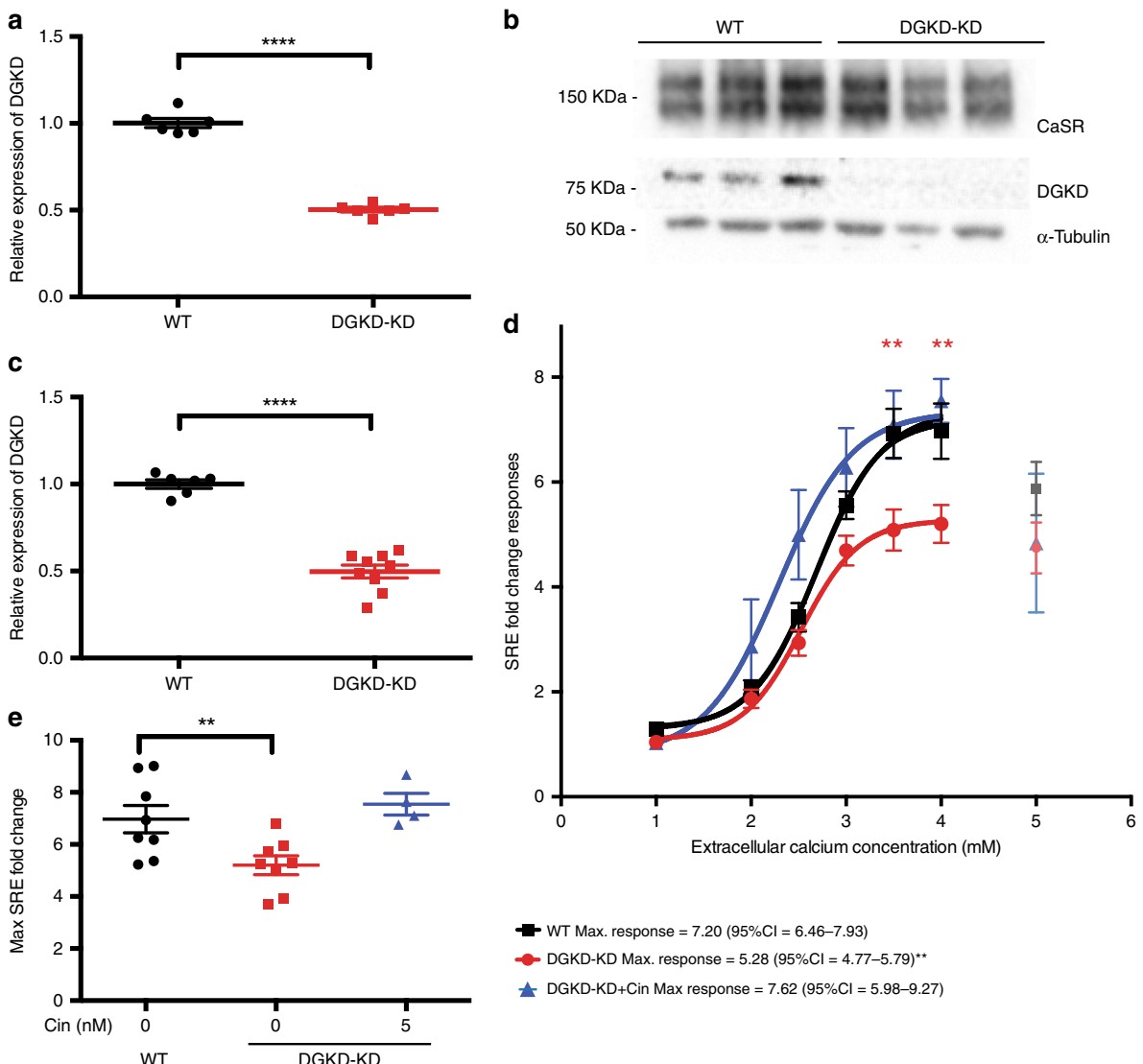

**Fig. 3** CaSR-mediated SRE responses following DGKD knockdown and effect of cinacalcet treatment in HEK-CaSR-SRE cells. **a** Relative expression of *DGKD*, as assessed by quantitative real-time PCR of HEK-CaSR-SRE cells treated with scrambled (WT) or *DGKD* (DGKD-KD) siRNA and used for SRE experiments. Samples were normalized to a geometric mean of four housekeeper genes: *PGK1*, *GAPDH*, *TUB1A*, *CDNK1B*. $n = 8$ biologically independent transfections. **b** Representative western blot of lysates from HEK-CaSR cells treated with scrambled or *DGKD* siRNA and used for SRE experiments. $\alpha$−Tubulin was used as a loading control. **c** Relative expression of DGKD, as assessed by densitometry of western blots from cells treated with scrambled or *DGKD* siRNA demonstrating a ~50% reduction in expression of DGKD following treatment with *DGKD* siRNA. Samples were normalized to PGK1. $n = 6$ biologically independent transfections for WT, $n = 9$ biologically independent transfections for DGKD-KD. **d** SRE responses of HEK-CaSR-SRE cells in response to changes in extracellular calcium concentration. Cells were treated with scrambled (WT) or *DGKD* (DGKD-KD) siRNA. The responses ± SEM are shown for $n = 8$ biologically independent transfections for WT and DGKD-KD cells and $n = 4$ biologically independent transfections for DGKD-KD + 5 nM cinacalcet cells. Treatment with *DGKD* siRNA led to a reduction in maximal response (red line) compared to cells treated with scrambled siRNA (black line). This loss-of-function could be rectified by treatment with 5 nM cinacalcet (blue line). Post desensitization points are shown but were not included in the analysis (gray, light red, and light blue). **e** Mean maximal responses with SEM of cells treated with scrambled siRNA (WT, black), *DGKD* siRNA (DGKD-KD, red) and *DGKD* siRNA incubated with 5 nM cinacalcet (blue). Statistical comparisons of maximal response were undertaken using F test. Student's t-tests were used to compare relative expression. Two-way ANOVA was used to compare points on dose response curve with reference to WT. Data are shown as mean ± SEM with **$p < 0.01$, ****$p < 0.0001$. Source data are provided as a Source Data file

3 standard deviations from the mean (calculated using UK Biobank's PCA-adjusted heterozygosity values, Data Field 20004). Individuals were then excluded who were not flagged by UK Biobank as having white British ancestry (on the basis of principal component analysis and self-reporting as "British" – Data Field 22006). Data was merged with publicly available data from the 1000 Genomes Project[45] and principal component analysis (PCA) performed using flashpca[46] to confirm that the white British ancestry individuals from UK Biobank overlapped with the "GBR" individuals from the 1000 Genomes Project. BOLT-LMM was used in analysis and therefore there were no sample exclusions based on relatedness[14]. In total, 86,693 individuals were excluded based on the above criteria. SNP-level QC was performed by excluding SNPs with Hardy–Weinberg equilibrium (HWE) $p < 10^{-4}$, < 98% call rate, and minor allele frequency (MAF) < 1%. Overall, 237,245 SNPs were excluded in total. Finally, six individuals were excluded who harboured an abnormal number of SNPs with a minor allele count of 1, or were visual outliers when autosomal heterozygosity was plotted against call rate. This resulted in a final dataset of 401,667 individuals and 547,011 SNPs. Following QC, individuals were excluded who were recorded to have a disorder of calcium homeostasis, malabsorption, or other condition known to predispose to kidney stone disease

(Supplementary Table 2). Subsequent case ascertainment was performed using the list of ICD-10 and OPCS codes for kidney and ureteric stones (Supplementary Table 1).

In the Japanese dataset samples were excluded if (i) call rate was < 0.98, (ii) they were from closely related individuals identified by identity-by-descent analysis, (iii) the samples were sex-mismatched with a lack of information, or (iv) they were non–East Asian outliers identified by principal component analysis of the studied samples and the three major reference populations (Africans, Europeans, and East Asians) in the International HapMap Project[47]. Standard quality-control criteria for variants were applied, excluding those with (i) SNP call rate < 0.99, (ii) minor allele frequency < 1%, and (iii) Hardy–Weinberg equilibrium $P$ value < $1.0 \times 10^{-6}$.

**Imputation**. UK Biobank phasing on the autosomes was performed using SHA-PEIT3[48], using the 1000 Genomes Phase 3 dataset as a reference panel. For imputation, both the HRC (Haplotype Reference Consortium) reference panel[49] and a merged UK10K/1000 Genomes Phase 3 panel were used. This resulted in a dataset with 92,693,895 autosomal SNPs, short indels and large structural variants. Imputation files were released in the BGEN (v1.2) file format[41].

In the case of Japanese data genotypes were prephased with MACH[50] and dosages imputed with minimac and the 1000 Genomes Project Phase 1 (version 3) East Asian reference haplotypes[45].

**Association analyses**. In the UK Biobank dataset genome-wide association analysis was undertaken across 547,011 genotyped SNPs and ~8.4 million imputed SNPs with MAF ≥ 0.01 and Info Score ≥ 0.9, using a linear mixed non-infinitesimal model implemented in BOLT-LMM v2.3[51]. A reference genetic map file for hg19 and a reference linkage disequilibrium (LD) score file for European-ancestry individuals included in the BOLT-LMM package in the analysis was used. Two covariates were used in the association study: genetic sex, and the genotyping platform (to account for array effects). Quantile–quantile and Manhattan plots were generated using FUMA.

The genomic inflation factor ($\lambda_{GC}$) and a value for the $\lambda_{GC}$ adjusted to a sample size of 1000 ($\lambda_{1000}$) are given for the UK Biobank GWAS, the BioBank Japan GWAS[11], and the trans-ethnic meta-analysis (Supplementary Table 10).

The LD score regression intercept[45] of 0.9997 with an attenuation ratio of 0.0075 indicated minimal inflation when adjusted for the large sample size. Conditional analysis at each associated locus was performed by conditioning on the allelic dosage (calculated using QCTOOL v2) of the most significantly associated SNP at each locus.

In the Japanese dataset GWAS was conducted using a logistic regression model by incorporating age, sex, and the top 10 principal components as covariates. Control individuals were younger and more commonly male; thus, these factors were included as covariates during association analysis.

Trans-ethnic meta-analysis was performed using the summary statistics from the UK and Japanese GWAS datasets (12,123 cases and 417,378 controls). An imputation quality score (RSQR) threshold of > 0.5 was applied to the Japanese GWAS SNPs[52] prior to performing a fixed-effects meta-analysis using GWAMA[53], using ~5 million SNPs common to both GWAS datasets. Quantile–quantile and Manhattan plots were generated using FUMA[19].

**In silico analyses**. The summary statistics from the GWAS meta-analysis were analyzed in FUMA[19] v1.3.3c, selecting UK Biobank Release 2 (White British) as the population reference panel (as the vast majority of individuals in the meta-analysis are of British rather than Japanese ethnicity). Functionally annotated SNPs were mapped to genes based on genomic position and annotations obtained from ANNOVAR, using positional mapping in FUMA (Supplementary Table 5)[54]. MAGMA (implemented in FUMA) was used to perform a gene-property analysis in order to identify particular tissue types relevant to kidney stones. This analysis determines if tissue-specific differential expression levels are predictive of the association of a gene with kidney stones, across 53 different tissues taken from the GTEx v7 database[55] (Supplementary Fig. 2).

In order to gain insight into the biological pathways implicated by the FUMA-prioritized genes, a gene set analysis was implemented using the GENE2FUNC tool in FUMA using the 54 positionally mapped genes with unique Entrez IDs and gene symbols. The following parameters were applied: Benjamini-Hochberg false discovery rate (FDR) for multiple testing correction, adjusted $p$-value cut-off = 0.0025, minimum number of overlapped genes = 2, GTEx v7 RNA-Seq expression data. Hypergeometric tests were performed to test if genes of interest are overrepresented in any of the pre-defined gene sets in GO biological processes (MsigDB v6.1) (Supplementary Fig. 3).

**Genotype–phenotype correlations**. Associations were sought between genotype and serum calcium (albumin adjusted), phosphate, parathyroid hormone, and 25-hydroxyvitamin D, and urinary calcium excretion and number of stone episodes. Patients were excluded from inclusion in genotype–phenotype correlations if they were known to have a disorder of calcium homeostasis, malabsorption, or other condition known to predispose to kidney stone disease.

**Characterization of the calcium-sensing receptor pathway**. Functional studies were undertaken using HEK293 cells (ATCC® CRL-1573™) that had been transfected to stably express calcium-sensing receptors (CaSRs) (HEK-CaSR cells). In addition, cells used for serum-response element (SRE) and nuclear factor or activated T-cells (NFAT) assays were stably transfected to express luciferase under the control of SRE (HEK-CaSR-SRE cells) or NFAT (HEK-CaSR-NFAT cells), respectively. Cells were transfected with 10 nM scrambled or *DGKD* siRNA (Qiagen) using lipofectamine RNAiMAX (Thermo Fisher Scientific) 72 h before experiments and maintained in DMEM-Glutamax media (Thermo Fisher Scientific) with 10% FBS (Gibco) and 400 μg/ml geneticin (Thermo Fisher Scientific) and 200 μg/ml hygromycin (Invitrogen) at 37 °C, 5% $CO_2$.

Successful knockdown of DGKD and maintenance of CaSR expression was confirmed via quantitative reverse transcriptase PCR (qRT-PCR) and western blot analyses. qRT-PCR analyses were performed in quadruplicate using Power SYBR Green Cells-to-CT™ Kit (Life Technologies), *DGKD, CASR, PGK1, GAPDH, TUB1A, CDNK1B* specific primers (Qiagen, Supplementary Table 11), and a Rotor-Gene Q real-time cycler (Qiagen Inc, Valencia, CA). Samples were normalized to a geometric mean of four housekeeper genes: *PGK1, GAPDH, TUB1A, CDNK1B*. Western blot analyses were undertaken using anti-DGKD (SAB1300472; Sigma; 1:1000), anti-CaSR (5C10, ADD; ab19347; Abcam; 1:1000), and anti–α–Tubulin (T5168; Sigma; 1:2000). The western blots were visualized using an Immuno-Star Western C kit (Bio-Rad) on a Bio-Rad Chemidoc XRS + system and relative expression of DGKD and CaSR were quantified by densitometry using ImageJ software. Both the 140 kDa immature CaSR band and the 160 kDa glycosylated band were considered in densitometry calculations. Captured images are included in the Source Data File.

To perform SRE response assays, at 60 h post transfection HEK-CaSR-SRE cells were incubated in 0.05% fetal bovine serum media with 0.45 mM calcium for 12 h, reducing extracellular calcium concentration and thus inducing basal cellular CaSR-mediated responses whilst maintaining cellular viability. At 72 h the media was changed to varying concentrations of extracellular calcium (0.1–5 mM), with either 5 nM cinacalcet or equivalent volume of DMSO (final concentration of DMSO 0.0001%), and the cells were incubated for a further 4 h at 37 °C. Cinacalcet (AMG-073 HCL) was obtained from Cambridge Bioscience (catalog CAY16042) and dissolved in DMSO prior to use in in vitro studies. Cells were lysed and luciferase activity measured using Luciferase Assay System (Promega) on a PHERAstar microplate reader (BMG Labtech). Assays were performed in > 4 biological replicates (independently transfected wells, performed on at least 4 different days). Nonlinear regression of concentration-response curves was performed with GraphPad Prism for determinations of maximal response.

To perform ERK phosphorylation assays HEK-CaSR cells were seeded in 96-well plates at 60 h post transfection. Cells were fasted for 4 h in 0.1 mM extracellular calcium then stimulated for 4 min with varying concentrations of extracellular calcium (0.1–5 mM), and lysed in Alphascreen Surefire lysis buffer. Alphascreen Surefire ERK phosphorylation assays were performed on whole cell lysates, as reported, and the fluorescence signal measured using a PHERAStar microplate reader (BMG Labtech)[56]. Assays were performed in > 4 biological replicates (independently transfected wells, performed on at least 4 different days). Nonlinear regression of concentration-response curves was performed with GraphPad Prism for determinations of maximal response.

To perform NFAT response assays, at 60 h post transfection HEK-CaSR-NFAT cells were incubated in 0.05% fetal bovine serum media with 0.45 mM calcium for 12 h, reducing extracellular calcium concentration and thus inducing basal cellular CaSR-mediated responses whilst maintaining cellular viability. At 72 h the media was changed to varying concentrations of extracellular calcium (0.1–5 mM) and the cells were incubated for a further 4 h at 37 °C. Cells were lysed and luciferase activity measured using Luciferase Assay System (Promega) on a PHERAstar microplate reader (BMG Labtech). Assays were performed in > 4 biological replicates (independently transfected wells, performed on at least 4 different days). Nonlinear regression of concentration-response curves was performed with GraphPad Prism for determinations of maximal response.

To measure intracellular calcium responses, at 60 h post transfection HEK-CaSR cells were plated in 96-well plates. At 70 h post transfection cells were washed with 100 μl Complete Imaging Buffer (150 mM NaCl, 2.6 mM KCl, 1.18 mM $MgCl_2$,10 mM HEPES, 0.1 mM $CaCl_2$, pH 7.4), loaded with Fluo-4 dye (Complete Imaging Buffer supplemented with; 1 μM Fluo-4 AM, 0.01% pluronic F-127, 0.5% BSA), and incubated for 60 min at 37 °C. Cells were washed again before the addition of a further 100 μl of Complete Imaging Buffer and incubated for 30 min at room temperature in the dark. Using an automated system, calcium chloride was injected into wells to achieve extracellular calcium concentrations ranging from 0.05 to 5 mM. Control wells were injected with 10 μM ionomycin. Changes in intracellular calcium concentrations were recorded via detection of fluorescence for 30 s using a PHERAstar microplate reader (BMG Labtech) at 37 °C with an excitation filter of 485 nm and an emission filter of 520 nm. The peak mean fluorescence ratio of the transient response following each individual stimulus was measured using MARS data analysis software (BMG Labtech). Relative fluorescence units were normalized to the fluorescence stimulated by ionomycin to account for differences in cell number and loading efficiency and then further normalized to the maximum response observed for the cells treated with scrambled siRNA[57,58]. Assays were performed using 4 biological replicates (independently transfected wells, performed on at least 4 different days). Nonlinear regression of

concentration–response curves was performed with GraphPad Prism for determinations of maximal response.

**Statistics**. For genotype-phenotype correlations, 2-tailed Student's *t*-tests were used for parametric data. Mann–Whitney U-tests were used for comparison of non-parametric data (number of stone episodes). ANOVA tests were used for comparisons of multiple sets of parametric data. Kruskall–Wallis tests were used for comparisons of multiple sets of non-parametric data (number of stone episodes). Significance was defined as $p < 0.05$ after Bonferroni correction for 7 tests on each set of data, thus $p < 0.05/7 = 0.007$. To calculate *p*-values for the additive effects of rs17216707 on serum calcium concentration and rs838717 on urinary calcium excretion, we performed a non-parametric test for trend across ordered groups (an extension of the Wilcoxon rank-sum test)[59], using the nptrend function in Stata[60]. For in vitro studies statistical comparisons were made with 2-tailed Student's *t*-tests, maximal responses were compared using the *F*-test, and responses at each extracellular calcium concentration were compared using a 2-way ANOVA with Tukey's multiple-comparisons test using GraphPad Prism 6[29,58].

**Reporting Summary**. Further information on research design is available in the Nature Research Reporting Summary linked to this article.

## Data availability

Full UK Biobank data are available by direct application to UK Biobank. Full GWAS summary statistics from the UK Biobank GWAS and the UK-Japanese meta-analysis can be found at: https://doi.org/10.5287/bodleian:2NEEgv2QD. Source data for Fig. 3, Table 2, and Supplementary Fig. 5 are included in the Source Data file. All other relevant data is available from the authors on request.

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

## Acknowledgements

This work was supported by grants from Kidney Research UK (RP_030_20180306) to S.A.H., A.W., M.G., B.W.T., and D.F, National Institute for Health Research (N.I.H.R) Oxford Biomedical Research Centre to R.V.T, and the Wellcome Trust (204826/z/16/z) to S.A.H, and M.G. S.A.H. is a N.I.H.R Academic Clinical Lecturer. A.W is an MRC Clinical Research Training Fellow. R.V.T. has Senior Investigator Awards from the Wellcome Trust (106995/z/15/z) and N.I.H.R. (NF-SI-0514–10091). We acknowledge the contribution to this study made by the Oxford Centre for Histopathology Research and the Oxford Radcliffe Biobank, which are supported by the NIHR Oxford Biomedical Research Centre.

## Author contributions

S.A.H., A.W., B.W.T. and D.F. designed this study. S.A.H., A.W., M.G., E.G., C.Ta., Y.K., C.Te., A.T., M.K., K.M. and B.W.T. acquired data. A.W. and D.F. carried out genetic association analysis. S.A.H., M.G., A.L.B., and A.K.G. undertook *in vitro* studies. S.A.H., A.W., M.G., M.N., C.Ta., Y.K., C.Te., A.T., M.K., K.M., R.V.T, B.W.T. and D.F. analyzed and interpreted data. S.A.H., A.W., R.V.T., B.W.T. and D.F. wrote the first draft of the manuscript. All other co-authors participated in the preparation of the manuscript by reading and commenting on the draft prior to submission.

## Competing interests

The authors declare no competing interests.

## Additional information

**Peer review information** *Nature Communications* thanks Bidhan Bandyopadhyay and other, anonymous, reviewers for their contributions to the peer review report of this work. Peer review reports are available.

