## [Peer Review File · Nature Communications]

Reviewers' Comments:

Reviewer #1:

Remarks to the Author:

The study reports genome-wide association findings for kidney stone trait in the UK Biobank white British (discovery) and combined meta-analyses with Biobank Japan resources. The main findings are the identification of 20 loci for kidney stone, including 10 novel. The authors then used a validation cohort of patients with kidney stones to examine the association of a SNP nearby the CYP24A1 gene with urinary/serum biomarkers of calcium metabolism, and performed in vitro studies of cells with reduced DGKD expression.

There is interest in understanding the genetic susceptibility to kidney stones to identify potential therapies for this highly prevalent disease. The study has several strengths including the use of large datasets for gene discovery and the additional exploration of gene effects in calcium metabolism using data from patients and through experiments. However, there are several aspects that need clarification and some important details buried in the supplementary material need to be added to the main text. Briefly, it is not clear which are the novel loci and if they were found in the UK biobank, Japan Biobank or in the European-Japanese analyses. In addition, there is likely phenotypic heterogeneity in the studies included in the discovery. It is unclear the relationship between the paper by Tanikawa et al (uploaded) and this manuscript. See full review below. Given the study includes European and East Asian populations, what is the relevance of kidney stone and its genetic determinants in these populations?

Line 57-58, what about the UMOD locus?

Phenotypic heterogeneity in case ascertainment in the UK biobank (ICD codes, potential single lifetime episode) and the Japan Biobank (patients seen in a kidney stone clinic likely with multiple episodes of kidney stone) needs to be discussed. How many of the clinical cases have a genetic diagnosis for monogenic disorders in calcium metabolism in the Japan Biobank? In addition, more details are needed on control samples in the Japan Biobank. Table S4 shows that cases were younger and predominantly men compared to controls, so it is unclear if controls reflect the population reference of cases. Some of this information needs to be included in the main text. Lines 75-76, I am not sure PAR is a good measure to report given the SNPs are imputed (which adds to the imprecision) and the PAR across SNPs is over 100%.

Gene selection for bioinformatics within GWAS loci was done using the approach "guilt by proximity". Is there other evidence for regulatory function of the variants?

Lines 96-97, related to the association of rs17216707 with calcium homeostasis in a cohort of patients with kidney stone: describe briefly the sample, if the calcium measures were obtained before treatment (drugs that can affect calcium, phosphorus including diuretics) and add this to the main text.

Line 98, Table 2 results for serum calcium are not supportive of a recessive effect given the increased serum calcium in participants with CC genotypes. Report p-values for additive effects so to be comparable with the GWAS discovery results.

Lines 102-104. Could this be because these patients were treated?

Lines 106-117: paragraph needs to be revised based on comments above.

Lines 151-155. Is the p-value for an additive genetic effect?

Conclusion: Are the findings in agreement with prior studies of monogenic disorders related to kidney stones? Are there new insights on calcium metabolism?

Line 389: what is the sample size for this validation study and were the biochemistry done before treatment.

Line 475. Explain why principal components were not used in models. Lambdas for UK biobank, Japan Biobank and European-Japanese meta-analyses need to be reported.

Table 1. add the n cases and controls for discovery and replication samples in a footnote. Add information on which loci are novel. Include information if the SNP is intronic, coding, or intergenic.

Table 2. Add total number of participants in the footnote.

Figure S1. Note of the plots shows the LD among most significant SNPs. The authors could include plots with European and East Asian LD.

Reviewer #2:

Remarks to the Author:

Table of Contents

1 Comments to the author

.. 1.1 minor comments

.. 1.2 major comments

1 Comments to the author

• In this report the authors perform a trans-ethnic genome-wide association meta-analysis for kidney stone disease utilizing large data sets from the UK Biobank and Japan. They identified novel loci, such as CYP24A1, and replicated previously reported loci (e.g. CaSR). The meta-analysis carefully performed, and well written. While these results are of interest and further add to the understanding of the genetic basis of kidney stone disease, I have some comments that would need clarification. The major concerns regard recessive and sex-specific effects of the reported variants and the usage of population attributable risk scores.

1.1 minor comments

• Table 1: The effect sizes between the UK Biobank and BioBank Japan seem to be very consistent judging from the confidence intervals of the odds ratios in Table 1. This is a sign of quality for a trans-ethnic meta-analysis, and it should be stated in the main text that no significant heterogeneity of effect is observed between the two data sets. A heterogeneity P-value could also be provided in Table 1 or in the supplementary material.

• Page 2, lines 31-32: It would be helpful if it was clearly stated which ten loci are considered as novel in regard to association to kidney stone disease. For example, as an extra column in Table 1. I would also like to point out that variants at WDR72 and POU2AF1 have recently been reported to associate with kidney stone disease by Benonisdottir et al. (PMID: 30476138). Variants at the UMOD locus on chromosome 16 have also been associated with kidney stone disease (Gudbjartsson et al. PMID: 20686651 and Oddsson et al. PMID: 26272126). The current report confirms the association of variants at these loci with kidney stone disease.

• Figure 1: For clarity, in Figure 1 B the association peaks could be labeled with gene symbols and color-coded as novel or reported.

- Page 4, lines 57-58: The sentence states that only two kidney stone disease associated loci have been replicated to date. On closer inspection of the literature it appears that two additional loci have been replicated. Oddsson et al. (PMID: 26272126) reported an association of variants at the ALPL locus with kidney stone disease and provided a replication in a Danish sample in the same study. Furthermore, a recent Chinese study also provided an independent replication of the association of the ALPL locus with kidney stones (PMID: 29489416). Also, in the report by Oddsson et al. a replication of the SLC34A1 and AQP1 loci reported by Urabe et al. was provided (PMID: 22396660). It appears that in total four loci associated with kidney stone disease have been replicated in populations of European and Asian descent.
- Page 7-8, line 58-61: For the sake of thoroughness, the number variants tested and significance thresholds used should be clearly stated at this point in the main text.
- Pages 5, lines 79-80: For clarity, the five loci identified to influence CaSR signaling should be clearly specified in the main text.
- Supplementary Table 1 and Supplementary Table 2: It would be informative to include the number individuals included in the study for each diagnostic code.

1.2 major comments

- Pages 4-5, lines 75-80: Population attributable risk is a measure of the impact of a sequence variant on a disease from a public health point of view. It is however a problematic measurement that is not the right one to evaluate the contribution of sequence variants to disease risk (see paper by Witte et al. for a discussion on the topic PMID: 25223781). For that purpose, another metric such as the sibling recurrent risk ratio might be more appropriate.
- Page 5, line 96-102:
 - To get a clearer picture of the potential recessive effect of the rs17216707 variant in CYP24A1 it would be of interest to:
 - calculate the genotypic odds ratios for kidney stone disease and test if the genotypic effect is consistent with a recessive model.
 - This is best achieved by testing a full genotypic model to estimate the effects of rs17216707[T] heterozygous and homozygous genotypes on kidney stone disease.
 - Significance levels for the heterozygous and homozygous genotypes can be retrieved by comparing the full model to homozygous and heterozygous models, respectively.
 - As this is an important question to address, the analysis would still be of interest if genotypic data is only available for one of the two data sets.

- It is problematic to conclude on an effect of a genotype when testing traits in a set of cases. If possible, it would be beneficial to also include non stone former controls into the analysis to get a to get a more accurate estimation of the underlying effects. Significance level, effects and confidence intervals for heterozygous and homozygous genotypes should be reported and can be calculated as described above.
- Page 7-8, line 151-159: To get a clearer picture of the potential difference of effect between the sexes for the rs838717 variant in DGKD it would be of interest to:
 - stratify the data by sex calculate the genotypic odds ratios for kidney stone disease and test if the genotypic effects are significantly different between the sexes.
 - As mentioned above, It is problematic to conclude on an effect of a genotype when testing traits in a set of cases. If possible, it would be beneficial to also include non stone former controls into the analysis to get a to get a more accurate estimation of the underlying effects. Significance level, effects and confidence intervals for heterozygous and homozygous genotypes should be reported and can be calculated as described above.

Reviewer #3:

Remarks to the Author:

In this manuscript, the authors reported their findings indicating that genetic test may be needed to identify individuals' risk in kidney stone formation. Authors suggest this as personalized medical care for stone disease towards the precision-medicine approach, by targeting CaSR-signaling or vitamin D activation pathways in patients with recurrent kidney stone former. The topic itself is valuable and the method based on genome-wide association studies and meta-analysis, seems useful for the treatment of a subset of patients with nephrolithiasis. However, there are some fundamental aspects of CaSR (GpCR) activation i.e. receptor desensitization and adaptation were not discussed. Moreover, in Figure 2 they did not mention about the CaSR signaling mediated activation Ca²⁺ entry pathway, perhaps by TRPC/SOCE; so, any dynamic Ca²⁺ mobilization pathway(s) was not considered in in-vitro functional experiments. Although, in a similar study, has already been reported by the authors (Cell Reports 22, 1054–1066), that CaSR mediated sustained signaling can occur by a non-canonical endosomal pathway, which they provide an explanation for the observed reduction in CaSR signaling because of AP2s mutation. CaSR have been shown as a part of Ca²⁺ channel signaling complex, expression of which increased at the plasma membrane due to CaSR agonist activation. Most importantly, the authors presented the experimental results, but did not attempt to clarify why the effect of CaSR can be diverse in different conditions. The mechanism behind the experimental results may be much more interesting.

Some more comments need to be addressed:

1. Authors did some in vitro expression study, but did not try to show how much influence of CaSR mediated signaling canonical vs non-canonical pathway.
2. It is not clear why the authors used HEK 293-stably expressed, stable expression sometime goes up and down depending on the condition, there are inducible system used for such purposes. Some study shows that CaSR in HEK293 cells stably expressing human CaSR (HEK-CaSR cells) varies when glucose concentration in the buffer was raised as glucose activate CSR.
3. Over-expression of CSR then siRNA? Native CSR may be good, endogenous method to show regulation can show a comparative situation in native cells.

4. CaSR western blot presented in Fig3B is questionable with multiple bands (no control), which needs to be quantified, or RT-PCR to show knockdown (with full gel). Moreover, this is not definitive test, they should add staining and/or gene expression data with WB and/or qRT-PCR to show CaSR and DGKD expression in WT and DGKD-KD.

5. Additional evidence is required to justify the assessment of intracellular MAPK responses. A suggestion would be to evaluate ERK phosphorylation in WT and DGKD-KD to assess downstream MAPK responses. Other downstream MAPK responses would also be appropriate here. Which is also important for evaluating downstream CaSR signaling?

Minor comments:

1. DMSO concs. were not mentioned as DMSO can also increase $[Ca^{2+}]_i$ release.
2. What is the rationale for using 0.05% FBS media with 0.45mM calcium for 12 hours.
3. In Results the figures referred as Fig. 3A-E, make sure that text and figures are matched as appropriate.

Resubmission of: Genetic variants of calcium and vitamin D metabolism in kidney stone disease by Howles et al

Reviewer #1 (Remarks to the Author):

The study reports genome-wide association findings for kidney stone trait in the UK Biobank white British (discovery) and combined meta-analyses with Biobank Japan resources. The main findings are the identification of 20 loci for kidney stone, including 10 novel. The authors then used a validation cohort of patients with kidney stones to examine the association of a SNP nearby the CYP24A1 gene with urinary/serum biomarkers of calcium metabolism, and performed in vitro studies of cells with reduced DGKD expression. There is interest in understanding the genetic susceptibility to kidney stones to identify potential therapies for this highly prevalent disease. The study has several strengths including the use of large datasets for gene discovery and the additional exploration of gene effects in calcium metabolism using data from patients and through experiments. However, there are several aspects that need clarification and some important details buried in the supplementary material need to be added to the main text. Briefly, it is not clear which are the novel loci and if they were found in the UK biobank, Japan Biobank or in the European-Japanese analyses.

We thank the reviewer for this helpful comment. To aid clarity we have modified both Table 1 and Figure 1 as shown below, marking the novel loci with an asterisk and in red, respectively.

Table 1. SNPs significantly associated with kidney stone disease at trans-ethnic meta-analysis

SNP Chr ^a	SNP	Position ^b	Annotation ^c	EA ^d	NEA ^e	Discovery GWAS in UK Biobank				Replication GWAS in BioBank Japan				Meta-Analysis		Candidate gene
						EAF ^f	INFO ^g	OR ^h	P ⁱ	EAF ^f	INFO ^g	OR ^h	P ⁱ	OR ^e	P ⁱ	
1	rs10917002	21836340	I	T	C	0.11	0.997	1.18 (1.12-1.25)	3.60×10⁻⁹	0.38	0.998	1.09 (1.04-1.15)	5.83×10 ⁻⁵	1.13 (1.09-1.17)	3.45×10⁻¹¹	ALPL
2	rs780093	27742603	I	T	C	0.38	1	1.08 (1.04-1.12)	3.60×10 ⁻⁵	0.56	0.997	1.14 (1.09-1.18)	1.10×10⁻⁸	1.10 (1.08-1.13)	1.31×10⁻¹³	GCKR
2	rs13003198*	234257105	IG	T	C	0.39	0.997	1.10 (1.06-1.14)	6.50×10 ⁻⁸	0.25	0.98	1.12 (1.06-1.18)	1.09×10 ⁻⁵	1.11 (1.07-1.14)	3.89×10⁻¹¹	DGKD
4	rs1481012*	89039082	I	G	A	0.11	0.994	1.12 (1.06-1.18)	4.30×10 ⁻⁵	0.30	0.994	1.11 (1.05-1.17)	1.50×10 ⁻⁵	1.11 (1.07-1.16)	2.79×10⁻⁸	ABCG2
5	rs56235845	176798040	S	G	T	0.33	0.986	1.16 (1.12-1.20)	9.10×10⁻¹⁵	0.31	0.87	1.18 (1.12-1.25)	1.88×10⁻¹¹	1.16 (1.13-1.20)	2.64×10⁻²¹	SLC34A1
6	rs1155347	39146230	IG	C	T	0.22	0.975	1.12 (1.07-1.17)	2.60×10 ⁻⁷	0.16	0.925	1.16 (1.08-1.24)	1.33×10 ⁻⁶	1.13 (1.09-1.17)	8.54×10⁻¹¹	KCNK5
6	rs77648599*	160624115	I	G	T	0.03	0.992	1.33 (1.21-1.47)	5.50×10⁻⁹	0.04	0.739	1.22 (1.06-1.44)	1.89×10 ⁻³	1.30 (1.20-1.42)	5.39×10⁻¹⁰	SLC22A2
7	rs12539707*	27626165	I	T	C	0.30	0.999	1.13 (1.08-1.17)	6.30×10⁻¹⁰	0.09	0.789	1.10 (1.01-1.21)	0.0268	1.12 (1.08-1.16)	1.09×10⁻¹⁰	HIBADH
7	rs12666466	30916430	I	G	C	0.03	0.994	1.22 (1.11-1.34)	5.00×10 ⁻⁵	0.12	0.989	1.17 (1.08-1.26)	2.80×10 ⁻⁶	1.19 (1.12-1.26)	3.26×10⁻⁸	AQP1
11	rs4529910	111243102	I	T	G	0.27	0.998	1.07 (1.02-1.11)	1.40×10 ⁻³	0.59	0.999	1.12 (1.08-1.16)	3.94×10 ⁻⁷	1.09 (1.06-1.12)	4.25×10⁻¹⁰	POU2AF
13	rs1037271	42779410	I	C	T	0.39	0.995	1.11 (1.07-1.15)	2.50×10⁻⁸	0.55	0.936	1.20 (1.15-1.24)	7.49×10⁻¹⁵	1.15 (1.12-1.18)	1.29×10⁻²⁴	DGKH
15	rs578595	53997089	I	C	A	0.46	0.996	1.09 (1.05-1.13)	2.50×10 ⁻⁶	0.69	0.996	1.11 (1.06-1.15)	2.25×10 ⁻⁵	1.09 (1.07-1.12)	6.26×10⁻¹¹	WDR72
16	rs77924615	20392332	I	A	G	0.20	0.980	1.13 (1.08-1.18)	1.80×10⁻⁸	0.22	0.984	1.17 (1.10-1.24)	2.80×10⁻⁹	1.14 (1.10-1.19)	1.14×10⁻¹³	UMOD
16	rs889299*	23381914	I	G	A	0.76	1	1.10 (1.05-1.14)	8.20×10 ⁻⁶	0.66	0.895	1.09 (1.04-1.14)	9.39×10 ⁻⁴	1.09 (1.06-1.13)	1.55×10⁻⁸	SCNN1B
17	rs1010269	59448945	I	G	A	0.83	0.981	1.08 (1.03-1.14)	7.10×10 ⁻⁴	0.56	0.87	1.17 (1.12-1.22)	4.82×10⁻¹¹	1.13 (1.10-1.17)	3.71×10⁻¹⁵	BCAS
17	rs4793434*	70352537	I	G	C	0.50	0.993	1.09 (1.05-1.13)	1.50×10 ⁻⁶	0.32	0.983	1.09 (1.04-1.15)	2.04×10 ⁻⁴	1.09 (1.06-1.12)	4.52×10⁻⁹	SOX9
19	rs3760702	14588237	IG	A	G	0.33	0.994	1.08 (1.05-1.13)	1.40×10 ⁻⁵	0.25	0.971	1.14 (1.08-1.20)	3.78×10 ⁻⁷	1.09 (1.07-1.13)	1.98×10⁻⁹	GIPCI
20	rs17216707	52732362	IG	T	C	0.81	0.961	1.17 (1.12-1.22)	9.90×10⁻¹²	0.92	0.766	1.24 (1.15-1.34)	5.90×10 ⁻⁶	1.19 (1.14-1.23)	7.82×10⁻¹⁸	CYP24A1
21	rs12626330	37835982	I	G	C	0.49	0.980	1.16 (1.12-1.20)	5.80×10⁻¹⁷	0.39	0.981	1.12 (1.07-1.18)	2.77×10 ⁻⁷	1.15 (1.12-1.18)	7.24×10⁻²¹	CLDN14
22	rs13054904*	23410918	I	A	T	0.26	0.999	1.15 (1.11-1.20)	3.30×10⁻¹²	0.02	0.967	1.05 (0.91-1.26)	0.505	1.14 (1.10-1.19)	4.49×10⁻¹²	BCR

^aChromosome. ^bBased on NCBI Genome Build 37 (hg19). ^cI denotes an intronic position, IG an intergenic position, and S a splice site position. ^dThe effect allele. ^eThe alternate (non-effect) allele. ^fThe effect allele frequency in the study population. ^gThe imputation quality score. ^hOdds ratio (95% confidence intervals). OR>1 indicative of increased risk with effect allele. ⁱP values less than the genome-wide significance threshold of 5.0×10⁻⁸ are shown in ***bold italics***. Discovery cohort 6,536 cases and 388,508 controls. Replication cohort 5,587 cases and 28,870 controls. *Loci not previously reported to associate with kidney stone disease in GWAS.

Figure 1. Results of trans-ethnic genome-wide association study in kidney stone disease. A trans-ethnic meta-analysis of kidney stone disease was performed for 12,123 patients with kidney stone disease and 416,928 controls from the UK Biobank and BioBank Japan. Panel A is a quantile-quantile plot of observed vs. expected p-values. The λ_{GC} demonstrated some inflation (1.0957), but the LD score regression (LDSC) intercept of 0.9997, with an attenuation ratio of 0.0075 indicated that the inflation was largely due to polygenicity and the large sample size. Panel B is a Manhattan plot showing the genome-wide p values ($-\log_{10}$) plotted against their respective positions on each of the autosomes. The horizontal red line shows the genome-wide significance threshold of 5.0×10^{-8} . Loci have been labelled with the primary candidate gene at each locus, as shown in Table 1. Novel GWAS-discovered kidney stone loci are highlighted in red.

In addition, there is likely phenotypic heterogeneity in the studies included in the discovery. It is unclear the relationship between the paper by Tanikawa et al (uploaded) and this manuscript. See full review below.

The paper by Tanikawa et al. that we previously submitted alongside our manuscript has now been published in the Journal of the American Society of Nephrology (<https://doi.org/10.1681/ASN.2018090942>). Therefore, the distinction between that paper and the current paper is now considerably clearer. Changes in the manuscript to reflect this include:

Line 64-66 – Main text

“Four genome-wide association studies of nephrolithiasis have been published identifying fifteen loci associated with disease⁷⁻¹⁰; however no trans-ethnic studies have been undertaken.”

Line 67-69 – Main text

“.....a genome-wide association study was undertaken using the UK Biobank resource¹¹ and subsequent meta-analysis performed with the summary statistics of the Biobank Japan nephrolithiasis genome-wide association study^{10,12}.....”

Given the study includes European and East Asian populations, what is the relevance of kidney stone and its genetic determinants in these populations?

We thank the reviewer for raising this important point. We have addressed this issue more fully in conjunction with the question of heterogeneity raised by Reviewer 2, below. We present the heterogeneity statistics and conclude that the genetic architecture of renal stone disease between the two populations is very similar:

Line 78-82 – Main text:

“The allelic effects were concordant across both studies at all 20 loci, with minimal evidence of heterogeneity between the two GWAS at the majority of loci, with a Q -statistic p -value > 0.05 at 17/20 loci (Supplementary Table 2), suggesting that the genetic architecture of kidney stone disease is very similar between populations of European and East Asian ancestry.”

Line 57-58, what about the UMOD locus?

We thank the reviewer for this helpful observation. We have now removed references to replicated loci and modified the text as follows:

Line 64-66 – Main text

“Four genome-wide association studies of nephrolithiasis have been published identifying fifteen loci associated with disease⁷⁻¹⁰; however no trans-ethnic studies have been undertaken.”

Phenotypic heterogeneity in case ascertainment in the UK biobank (ICD codes, potential single lifetime episode) and the Japan Biobank (patients seen in a kidney stone clinic likely with multiple episodes of kidney stone) needs to be discussed. How many of the clinical cases have a genetic diagnosis for monogenic disorders in calcium metabolism in the Japan Biobank? In addition, more details are needed on control samples in the Japan Biobank. Table S4 shows that cases were younger and predominantly men compared to controls, so it is unclear if controls reflect the population reference of cases. Some of this information needs to be included in the main text.

We thank for the reviewer for this comment. There is indeed phenotypic heterogeneity between case ascertainment in the UK biobank and the Japan Biobank. Unfortunately, detailed clinical data, including diagnoses of monogenetic disorders in calcium metabolism was not available through the Japan Biobank. To highlight this, we have included the following statement in the online methods:

Line 758-659 – online methods:

“Information regarding conditions known to predispose to kidney stones was unavailable.”

In addition, more details are needed on control samples in the Japan Biobank. Table S4 shows that cases were younger and predominantly men compared to controls, so it is unclear if controls reflect the population reference of cases. Some of this information needs to be included in the main text.

We thank for the reviewer for this comment. The control individuals are indeed younger and more commonly male than those identified as cases. In light of this, age and sex were included as covariates during association analysis. To highlight this the following text has been added to the manuscript:

Line 850-852 – online methods:

“Control individuals were younger and more commonly male; thus, these factors were included as covariates during association analysis.”

Lines 75-76, I am not sure PAR is a good measure to report given the SNPs are imputed (which adds to the imprecision) and the PAR across SNPs is over 100%.

We accept the criticism of the use of PAR, raised by both Reviewer 1 and Reviewer 2. We have therefore removed these analyses from the manuscript.

Gene selection for bioinformatics within GWAS loci was done using the approach “guilt by proximity”. Is there other evidence for regulatory function of the variants?

We thank the reviewer for this comment. With regard to “guilt by proximity” We used FUMA (Watanabe et al., Nat Comms, 2017, PMID: 29184056) to map candidate genes to the associated loci. As stated in the FUMA authors’ paper, there are two components to positional mapping. Firstly, SNPs are annotated with their biological functionality, including ANNOVAR scores, deleteriousness scores (CADD), and potential regulatory function (RegulomeDB score). FUMA then maps these SNPs to genes based on potential functional consequences and their physical position in the genome.

As such, the candidate genes that we have listed are a product not only of physical proximity to the SNPs, but also of potential function. We employed similar positional mapping to implicate genes in our recent GWAS of carpal tunnel syndrome (Wiberg et al. Nat Comms 2019, PMID: 30833571).

With regard to evidence for regulatory function of the variants, we have created a new table (Supplementary Data 1), which shows the meta-analysis significant SNPs with likely functional consequences, based on RegulomeDB and CADD scores, and we have amended the text accordingly:

Line 82-101 – Main text:

“Out of 849 SNPs with a meta-analysis significant p-value of $p < 5.0 \times 10^{-8}$, many demonstrated evidence of potential functionality: 33 SNPs had a combined annotation-dependent depletion (CADD) score > 12.37 , the threshold suggested for deleterious SNPs¹³. A further 54 SNPs had a RegulomeDB score of 2b or higher, which is likely to affect protein binding¹⁴ (Supplementary Data 1).”

Lines 96-97, related to the association of rs17216707 with calcium homeostasis in a cohort of patients with kidney stone: describe briefly the sample, if the calcium measures were obtained before treatment (drugs that can affect calcium, phosphorus including diuretics) and add this to the main text.

We thank the reviewer for this comment, a brief description of the cohort has been added and an analysis of drug intake has been undertaken, which has shown that intake of medications that may affect calcium and phosphorous were comparable across genotypes:

Line 118-122 – Main text:

“associations of rs17216707were sought in a validation cohort of 440 kidney stone formers attending the Oxford University Hospitals NHS Foundation Trust for treatment of kidney stones.”

Line 151-153 – Main text:

“Intake of medications including steroids and diuretics was comparable across genotypes (Supplementary Fig. 4A).”

Supplementary Figure 4:

Supplementary Figure 4: Intake of medications in validation cohort participants shown across genotypes (oral steroids, diuretics, and medications for indigestion).

Line 98, Table 2 results for serum calcium are not supportive of a recessive effect given the increased serum calcium in participants with CC genotypes. Report p-values for additive effects so to be comparable with the GWAS discovery results.

We agree with the reviewer that the evidence for this is equivocal and have therefore removed the words “consistent with a recessive effect” from the text. We also present a p-value for an additive effect based on a trend test, as suggested by the reviewer. However, our analyses of genotypic odds ratios suggested by Reviewer 2 (below) does suggest the strongest evidence for a recessive model (over an additive or dominant model), so we do allude to this in the sentence that follows (please see our response to Reviewer 2 for full details of this analysis):

Line 141-146 – Main text:

“Individuals homozygous for the CYP24A1 increased-risk allele rs17216707 (T) had a significantly increased mean serum calcium concentration when compared to heterozygotes (mean serum calcium 2.36mmol/l (TT) vs. 2.32mmol/l (TC); trend test p-value for additive effect = 0.023) (Table 2). Analysis of the allelic frequencies of this SNP between cases and controls within the UK Biobank cohort provided strongest evidence in favour of a recessive model (Supplementary Table 4).”

Line 694-695 - Table 2 Figure Legend:

“Trend tests were performed for additive effects of rs17216707 on serum calcium ($p=0.023$) and rs838717 on urinary calcium excretion ($p=0.017$).”

Line 968 - 971 - Online Methods:

“To calculate p -values for the additive effects of rs17216707 on serum calcium concentration and rs838717 on urinary calcium excretion, we performed a non-parametric test for trend across ordered groups (an extension of the Wilcoxon rank-sum test)⁵⁶, using the `np trend` function in Stata⁵⁷”

Lines 102-104. Could this be because these patients were treated?

We thank the reviewer for this comment, patients with disorders of calcium homeostasis were excluded from the analysis and the intake of medications including steroids and diuretics was comparable across genotypes. This is emphasized in the following text:

Line 883-885 – Online Methods:

“Patients were excluded from inclusion in genotype-phenotype correlations if they were known to have a disorder of calcium homeostasis, malabsorption, or other condition known to predispose to kidney stone disease.”

Line 251-253 – Main text:

“Prescription of medications, including steroids and diuretics, was comparable across DGKD and DGKH genotypes (Supplementary Fig. 4B and 4C).”

Supplementary Figure 4:

Supplementary Figure 4: Intake of medications in validation cohort participants shown across genotypes (oral steroids, diuretics, and medications for indigestion).

Lines 106-117: paragraph needs to be revised based on comments above.

We thank the reviewer for this suggestion, we have incorporated changes to the text as outlined above. However, these changes and comments do not affect the conclusions that we reach in paragraph 106-117, this text has therefore been left unaltered.

Lines 151-155. Is the p-value for an additive genetic effect?

The previously reported p-value of 0.0055 referred to the t-test between the GG and AA genotypes at this SNP. However, we now present a trend test p-value for an additive effect, as per the Reviewer's suggestion to report an additive p-value for rs17216707 (above):

Line 218-223 – Main text:

“The DGKD increased-risk allele rs838717 (G) (top DGKD-associated SNP in the UK Biobank GWAS, Supplementary Table 1, linkage disequilibrium with rs13003198 $r^2=0.53$) associated with increased 24-hour urinary calcium excretion in male stone formers (mean 24-hour urinary calcium excretion 7.27mmol (GG) vs. 4.54mmol (AA); trend test p-value for additive effect=0.017) (Table 2), consistent with enhanced CaSR-signal transduction.”

Conclusion: Are the findings in agreement with prior studies of monogenic disorders related to kidney stones?

We thank the reviewer for this comment, our findings highlight the role of CYP24A1 as a factor in the pathogenesis of kidney stone disease, this is in agreement with its role in the monogenic disorder of infantile hypercalcaemia type 1. In addition, our study sheds light on the role of enhanced calcium-sensing receptor signaling in common forms of hypercalciuria in addition to the rare monogenetic disorder of autosomal dominant hypocalcaemia. We have brought attention to these observations as follows:

Line 155-160 – Main text:

“These findings support our hypothesis that the rs17216707 increased-risk allele is associated with a relative hypercalcaemia and reduced activity of the 24-hydroxylase enzyme and highlight the role of vitamin D catabolism in kidney stone formation. Patients with loss-of-function CYP24A1 mutations with hypercalcaemia and recurrent kidney stone disease have been successfully treated with inhibitors of vitamin D synthesis including fluconazole, similar therapies may be useful in rs17216707 (TT) recurrent kidney stone formers¹⁹.”

Line 207-212 – Main text:

“Gain-of-function mutations in components of the CaSR-signaling pathway result in autosomal dominant hypocalcaemia (ADH, OMIM 601198, 615361), which is associated with hypercalciuria in ~10% of individuals^{27,28}. ADH-associated mutations result in a gain-of-function in CaSR-intracellular signaling in vitro via pathways including intracellular calcium ions or MAPK^{27,29,30}.”

Are there new insights on calcium metabolism?

We thank the reviewer for this comment. We believe that this study does bring new insights on calcium metabolism in particular the role of variable vitamin D sensitivity in calcium metabolism and CaSR-signaling in regulation of urinary calcium excretion. Of note, since the initial submission, we have undertaken additional *in vitro* studies that have demonstrated that whilst CaSR-signaling via the MAPK pathway is affected by DGKD knockdown, signalling via the intracellular calcium pathway is unaffected, as outlined in the text below:

Line 255-301 – Main text:

“To investigate the role of DGKD in CaSR-signalling via the MAPK pathway, HEK-CaSR-SRE and HEK-CaSR cells were treated with scrambled or DGKD targeted siRNA and intracellular MAPK responses to alterations in extracellular calcium concentration assessed via SRE and ERK-phosphorylation (pERK) assays, respectively. Treatment with DGKD targeted siRNA resulted in a reduction in DGKD expression when compared to cells treated with scrambled siRNA without alteration in CaSR expression (Fig. 3A-C and Supplementary Fig. 5A and B). SRE and pERK responses were significantly decreased in cells with reduced DGKD expression (DGKD-KD) when compared to cells with baseline DGKD expression (WT) (SRE maximal response DGKD-KD=5.28 fold change, 95%

confidence interval (CI)=4.77-5.79 vs. WT=7.20 fold change, 95% CI=6.46-7.93, $p=0.0065$; pERK maximal response DGKD-KD=24.77, 95% CI=22.16-27.38 vs. WT=39.46 fold change, 95% CI=34.07-44.84, $p=0.0056$). Cinacalcet rectified this loss-of-function in SRE-reporter assays (DGKD-KD+5nM cinacalcet, maximal response=7.62 fold change, 95% CI=5.98-9.27) (Fig. 3D-E and Supplementary Fig. 5C).

Furthermore, to investigate the role of DGKD in CaSR-signaling via intracellular calcium mobilization, HEK-CaSR-NFAT and HEK-CaSR cells were treated with scrambled or DGKD targeted siRNA and intracellular NFAT and intracellular calcium responses to alterations in extracellular calcium concentration assessed via NFAT-reporter and Fluo-4 calcium assays, respectively. Intracellular calcium responses were unaffected by a reduction in DGKD expression (NFAT maximal response DGKD-KD=2.91 fold change, 95% CI=2.69-3.12 vs. WT=2.85 fold change, 95% CI=2.62-3.08, $p=0.73$; Fluo-4 maximal response DGKD-KD=96.16, 95% CI=89.83-102.5 vs. WT=96.78 fold change, 95% CI=92.76-100.8, $p=0.83$) (Supplementary Fig. 5D and E).

These findings provide evidence that DGKD influences CaSR-mediated signal transduction and suggest that the DGKD increased-risk allele may associate with a relative increase in DGKD expression thereby enhancing CaSR-mediated signaling via the MAPK pathway whilst leaving signaling via the intracellular calcium pathway unaffected. This biased signaling may provide an explanation for the observed correlation of the DGKD increased-risk allele rs838717 (G) with increased urinary calcium excretion but not serum calcium concentration (Table 2). Calcilytics, including NPS-2143 and ronacaleret, rectify enhanced CaSR-mediated signaling in vitro and biochemical phenotypes in mouse models of ADH³⁰⁻³². We predict that the development of biased calcilytics may provide a novel, targeted therapeutic approach to reduce urinary calcium excretion in recurrent stone formers carrying CaSR-associated increased-risk alleles.”

The main text also highlights our findings as follows:

Line 316-317 – Main text:

“...and revealed the importance of vitamin D metabolism pathways and enhanced and biased CaSR-signaling in the pathogenesis of nephrolithiasis”

Line 389: what is the sample size for this validation study and were the biochemistry done before treatment.

We thank the reviewer for this comment, we have added the sample size to the text as follows:

Line 665-666 – Online Methods:

“Four hundred and forty patients attending the Oxford University Hospitals NHS Foundation Trust for treatment of kidney stones were enrolled....”

In addition, we have studied drug intake across genotypes and found this to be comparable:

Line 151-153 – Main text:

“Intake of medications including steroids and diuretics was comparable across genotypes (Supplementary Fig. 4A)..”

Line 251-253 – Main text:

“Prescription of medications, including steroids and diuretics, was comparable across DGKD and DGKH genotypes (Supplementary Fig. 4B and 4C).”

Line 475. Explain why principal components were not used in models. Lambdas for UK biobank, Japan Biobank and European-Japanese meta-analyses need to be reported.

Re: principal components:

The association study was undertaken using a linear mixed non-infinitesimal model implemented in BOLT-LMM. Such mixed-model association methods have become a widely adopted method for performing association analyses for very large datasets such as UK Biobank. The principal advantage conferred by this method (over more traditional linear regression methods) is that it is robust to potential confounding due to population structure such as relatedness (meaning that individuals need not be excluded on the basis of kinship).

In a description of the methods, the authors of the BOLT-LMM software (Loh et al., Nat Genet 2018, PMID: 29892013) state that principal components can be used as covariates for the purpose of accelerating the computations. However, the authors also make it clear that principal components do not alter the result output, and are thus not deemed necessary, which is why we proceeded without using them.

Re: Lambdas

We now report the lambdas (along with lambda1000) for the three studies as Supplementary Table 9, and have added a sentence in the Online Methods to make this clear:

Line 842-844 – Online Methods:

“The genomic inflation factor (λ_{GC}) and a value for the λ_{GC} adjusted to a sample size of 1000 (λ_{1000}) are given for the UK Biobank GWAS, the BioBank Japan GWAS¹⁰, and the trans-ethnic meta-analysis (Supplementary Table 9).”

Supplementary Materials:

Supplementary Table 9. Genomic inflation in the GWA studies. The genomic inflation factor (λ_{GC}) and a value for the λ_{GC} adjusted to a sample size of 1000 (λ_{1000}) are given for the UK Biobank GWAS, the Japanese GWAS, and the trans-ethnic meta-analysis.

Study	λ_{GC}	λ_{1000}
UK Biobank	1.096	1.030
BioBank Japan	1.164	1.008
Trans-ethnic meta-analysis	1.125	1.021

Table 1. add the n cases and controls for discovery and replication samples in a footnote. Add information on which loci are novel.

We thank the reviewer for this suggestion. These additions have been made:

Line 672 - Table 1 – Main text:

*“Discovery cohort 6,536 cases and 388,508 controls. Replication cohort 5,587 cases and 28,870 controls. *Loci not previously reported to associate with kidney stone disease at GWAS.”*

Table 2. Add total number of participants in the footnote.

We thank the reviewer for this suggestion. This addition has been made:

Line 691 - Table 2 – Main text:

“A total of 440 patients were recruited...”

Figure S1. Note of the plots shows the LD among most significant SNPs. The authors could include plots with European and East Asian LD.

We thank the Reviewer for this suggestion. We have amended this figure accordingly, and for each SNP, we now show two plots – one with European LD relationships, and the other with East Asian LD relationships. An example for the first SNP is shown below:

Supplementary Materials - Figure 1

Supplementary Figure 1. Regional plots of all GWAS-associated loci in the UK-Japanese meta-analysis. LocusZoom plots for the 20 index SNPs are shown, ordered by chromosome number and genomic position. For each SNP, one plot shows the r^2 relationship between the SNPs based on LD relationships in European populations (hg19/1000 Genomes Nov 2014 EUR) and the other in East Asian populations (hg19/1000 Genomes Nov 2014 ASN). SNP position is shown on the x-axis, and strength of association on the y-axis. Genes within 500 kb of the index SNP are shown in the lower panel.

Reviewer #2 (Remarks to the Author):

Table of Contents

1 Comments to the author
.. 1.1 minor comments
.. 1.2 major comments

1 Comments to the author

• In this report the authors perform a trans-ethnic genome-wide association meta-analysis for kidney stone disease utilizing large data sets from the UK Biobank and Japan. They identified novel loci, such as CYP24A1, and replicated previously reported loci (e.g. CaSR). The meta-analysis carefully performed, and well written. While these results are of interest and further add to the understanding of the genetic basis of kidney stone disease, I have some comments that would need clarification. The major concerns regard recessive and sex-specific effects of the reported variants and the usage of population attributable risk scores.

1.1 minor comments

• Table 1: The effect sizes between the UK Biobank and BioBank Japan seem to be very consistent judging from the confidence intervals of the odds ratios in Table 1. This is a sign of quality for a trans-ethnic meta-analysis, and it should be stated in the main text that no significant heterogeneity of effect is observed between the two data sets. A heterogeneity P-value could also be provided in Table 1 or in the supplementary material.

As the reviewer notes, there was little heterogeneity in effect sizes between the UK Biobank and Japanese GWAS. We have calculated the heterogeneity q-statistics and their p-values from the meta-analysis, and present these as Supplementary Table 2:

Supplementary Table 2: Heterogeneity in effects of index SNPs between UK Biobank GWAS and Japanese GWAS. The *Q*-statistics and corresponding *p*-values are shown for each of the 20 index SNPs that were genome-wide significant in the trans-ethnic meta-analysis. Out of 20 loci, 3 loci demonstrated heterogeneity between the two GWAS at $p < 0.05$ (bold italic).

Chromosome	SNP ID	Q statistic	Q statistic p-value
1	rs10917002	4.179	0.041
2	rs780093	3.853	0.050
2	rs13003198	0.127	0.721
4	rs1481012	0.079	0.779
5	rs56235845	0.346	0.556
6	rs1155347	0.659	0.417
6	rs77648599	0.840	0.359
7	rs12539707	0.234	0.628
7	rs12666466	0.506	0.477
11	rs4529910	3.133	0.077
13	rs1037271	8.465	0.004
15	rs578595	0.419	0.518
16	rs77924615	0.513	0.474
16	rs889299	0.104	0.747
17	rs1010269	5.750	0.016
17	rs4793434	0.003	0.957
19	rs3760702	1.697	0.193
20	rs17216707	1.846	0.174
21	rs12626330	1.092	0.296
22	rs13054904	0.893	0.345

We now also allude to this low overall heterogeneity in the main manuscript:

Line 78-82 – Main text:

“The allelic effects were concordant across both studies at all 20 loci, with minimal evidence of heterogeneity between the two GWAS at the majority of loci, with a Q-statistic p-value > 0.05 at 17/20 loci (Supplementary Table 2), suggesting that the genetic architecture of kidney stone disease is very similar between populations of European and East Asian ancestry.”

• Page 2, lines 31-32: It would be helpful if it was clearly stated which ten loci are considered as novel in regard to association to kidney stone disease. For example, as an extra column in Table 1. I would also like to point out that variants at WDR72 and POU2AF1 have recently been reported to associate with kidney stone disease by Benonisdottir et al. (PMID: 30476138). Variants at the UMOD locus on chromosome 16 have also been associated with kidney stone disease (Gudbjartsson et al. PMID: 20686651 and Oddsson et al. PMID: 26272126). The current report confirms the association of variants at these loci with kidney stone disease.

We thank the reviewer for this comment and highlighting the recent manuscript by Benonisdottir et al. Now that the paper by Tanikawa et al. that we previously submitted alongside our manuscript has been published in the Journal of the American Society of Nephrology (<https://doi.org/10.1681/ASN.2018090942>), we would consider seven of the loci to be novel and have modified Table 1 to highlight these:

Table 1. SNPs significantly associated with kidney stone disease at trans-ethnic meta-analysis

Chr ^a	SNP	SNP				Discovery GWAS in UK Biobank				Replication GWAS in BioBank Japan				Meta-Analysis		Candidate gene
		Position ^b	Annotation ^c	EA ^d	NEA ^e	EAF ^f	INFO ^g	OR ^h	P ⁱ	EAF ^f	INFO ^g	OR ^h	P ⁱ	OR ^g	P ⁱ	
1	rs10917002	21836340	I	T	C	0.11	0.997	1.18 (1.12-1.25)	3.60×10⁻⁹	0.38	0.998	1.09 (1.04-1.15)	5.83×10 ⁻⁵	1.13 (1.09-1.17)	3.45×10⁻¹¹	ALPL
2	rs780093	27742603	I	T	C	0.38	1	1.08 (1.04-1.12)	3.60×10 ⁻⁵	0.56	0.997	1.14 (1.09-1.18)	1.10×10⁻⁸	1.10 (1.08-1.13)	1.31×10⁻¹³	GCKR
2	rs13003198*	234257105	IG	T	C	0.39	0.997	1.10 (1.06-1.14)	6.50×10 ⁻⁸	0.25	0.98	1.12 (1.06-1.18)	1.09×10 ⁻⁵	1.11 (1.07-1.14)	3.89×10⁻¹¹	DGKD
4	rs1481012*	89039082	I	G	A	0.11	0.994	1.12 (1.06-1.18)	4.30×10 ⁻⁵	0.30	0.994	1.11 (1.05-1.17)	1.50×10 ⁻⁵	1.11 (1.07-1.16)	2.79×10⁻⁸	ABCG2
5	rs56235845	176798040	S	G	T	0.33	0.986	1.16 (1.12-1.20)	9.10×10⁻¹⁵	0.31	0.87	1.18 (1.12-1.25)	1.88×10⁻¹¹	1.16 (1.13-1.20)	2.64×10⁻²¹	SLC34A1
6	rs1155347	39146230	IG	C	T	0.22	0.975	1.12 (1.07-1.17)	2.60×10 ⁻⁷	0.16	0.925	1.16 (1.08-1.24)	1.33×10 ⁻⁶	1.13 (1.09-1.17)	8.54×10⁻¹¹	KCNK5
6	rs77648599*	160624115	I	G	T	0.03	0.992	1.33 (1.21-1.47)	5.50×10⁻⁹	0.04	0.739	1.22 (1.06-1.44)	1.89×10 ⁻³	1.30 (1.20-1.42)	5.39×10⁻¹⁰	SLC22A2
7	rs12539707*	27626165	I	T	C	0.30	0.999	1.13 (1.08-1.17)	6.30×10⁻¹⁰	0.09	0.789	1.10 (1.01-1.21)	0.0268	1.12 (1.08-1.16)	1.09×10⁻¹⁰	HIBADH
7	rs12666466	30916430	I	G	C	0.03	0.994	1.22 (1.11-1.34)	5.00×10 ⁻⁵	0.12	0.989	1.17 (1.08-1.26)	2.80×10 ⁻⁶	1.19 (1.12-1.26)	3.26×10⁻⁸	AQP1
11	rs4529910	111243102	I	T	G	0.27	0.998	1.07 (1.02-1.11)	1.40×10 ⁻³	0.59	0.999	1.12 (1.08-1.16)	3.94×10 ⁻⁷	1.09 (1.06-1.12)	4.25×10⁻¹⁰	POU2AF
13	rs1037271	42779410	I	C	T	0.39	0.995	1.11 (1.07-1.15)	2.50×10⁻⁸	0.55	0.936	1.20 (1.15-1.24)	7.49×10⁻¹⁵	1.15 (1.12-1.18)	1.29×10⁻²⁴	DGKH
15	rs578595	53997089	I	C	A	0.46	0.996	1.09 (1.05-1.13)	2.50×10 ⁻⁶	0.69	0.996	1.11 (1.06-1.15)	2.25×10 ⁻⁵	1.09 (1.07-1.12)	6.26×10⁻¹¹	WDR72
16	rs77924615	20392332	I	A	G	0.20	0.980	1.13 (1.08-1.18)	1.80×10⁻⁸	0.22	0.984	1.17 (1.10-1.24)	2.80×10⁻⁹	1.14 (1.10-1.19)	1.14×10⁻¹³	UMOD
16	rs889299	23381914	I	G	A	0.76	1	1.10 (1.05-1.14)	8.20×10 ⁻⁶	0.66	0.895	1.09 (1.04-1.14)	9.39×10 ⁻⁴	1.09 (1.06-1.13)	1.55×10⁻⁸	SCNN1B
17	rs1010269	59448945	I	G	A	0.83	0.981	1.08 (1.03-1.14)	7.10×10 ⁻⁴	0.56	0.87	1.17 (1.12-1.22)	4.82×10⁻¹¹	1.13 (1.10-1.17)	3.71×10⁻¹⁵	BCAS
17	rs4793434*	70352537	I	G	C	0.50	0.993	1.09 (1.05-1.13)	1.50×10 ⁻⁶	0.32	0.983	1.09 (1.04-1.15)	2.04×10 ⁻⁴	1.09 (1.06-1.12)	4.52×10⁻⁹	SOX9
19	rs3760702	14588237	IG	A	G	0.33	0.994	1.08 (1.05-1.13)	1.40×10 ⁻⁵	0.25	0.971	1.14 (1.08-1.20)	3.78×10 ⁻⁷	1.09 (1.07-1.13)	1.98×10⁻⁹	GIPCI
20	rs17216707	52732362	IG	T	C	0.81	0.961	1.17 (1.12-1.22)	9.90×10⁻¹²	0.92	0.766	1.24 (1.15-1.34)	5.90×10 ⁻⁶	1.19 (1.14-1.23)	7.82×10⁻¹⁸	CYP24A1
21	rs12626330	37835982	I	G	C	0.49	0.980	1.16 (1.12-1.20)	5.80×10⁻¹⁷	0.39	0.981	1.12 (1.07-1.18)	2.77×10 ⁻⁷	1.15 (1.12-1.18)	7.24×10⁻²¹	CLDN14
22	rs13054904*	23410918	I	A	T	0.26	0.999	1.15 (1.11-1.20)	3.30×10⁻¹²	0.02	0.967	1.05 (0.91-1.26)	0.505	1.14 (1.10-1.19)	4.49×10⁻¹²	BCR

^aChromosome. ^bBased on NCBI Genome Build 37 (hg19). ^cI denotes an intronic position, IG an intergenic position, and S a splice site position. ^dThe effect allele. ^eThe alternate (non-effect) allele. ^fThe effect allele frequency in the study population. ^gThe imputation quality score. ^hOdds ratio (95% confidence intervals). OR>1 indicative of increased risk with effect allele. ⁱP values less than the genome-wide significance threshold of 5.0×10⁻⁸ are shown in **bold italics**. Discovery cohort 6,536 cases and 388,508 controls. Replication cohort 5,587 cases and 28,870 controls. *Loci not previously reported to associate with kidney stone disease in GWAS.

In addition, we have modified the main text and made reference to Benonisdottir et al, Gudbjartsson et al. and Oddsson et al. as follows:

Line 64-66 – Main text:

“Four genome-wide association studies of nephrolithiasis have been published identifying fifteen loci associated with disease⁷⁻¹⁰; however no trans-ethnic studies have been undertaken.”

Line 76-78 – Main text:

“Seven of the identified loci have not previously been reported to associate with kidney stone disease at GWAS (Table 1)^{7-10,13,14}.”

- Figure 1: For clarity, in Figure 1B the association peaks could be labeled with gene symbols and color-coded as novel or reported.

We thank the reviewer for this suggestion. We have amended this figure accordingly, and colour-coded the gene names to make clear which loci are novel.

Figure 1. Results of trans-ethnic genome-wide association study in kidney stone disease. A trans-ethnic meta-analysis of kidney stone disease was performed for 12,123 patients with kidney stone disease and 416,928 controls from the UK Biobank and BioBank Japan. Panel A is a quantile-quantile plot of observed vs. expected p-values. The λ_{GC} demonstrated some inflation (1.0957), but the LD score regression (LDSC) intercept of 0.9997, with an attenuation ratio of 0.0075 indicated that the inflation was largely due to polygenicity and the large sample size. Panel B is a Manhattan plot showing the genome-wide p values ($-\log_{10}$) plotted against their respective positions on each of the autosomes. The horizontal red line shows the genome-wide significance threshold of 5.0×10^{-8} . Loci have been labelled with the primary candidate gene at each locus, as shown in Table 1. Novel GWAS-discovered kidney stone loci are highlighted in red.

- Page 4, lines 57-58: The sentence states that only two kidney stone disease associated loci have been replicated to date. On closer inspection of the literature it appears that two additional loci have been replicated. Oddsson et al. (PMID: 26272126) reported an association of variants at the ALPL locus with kidney stone disease and provided a replication in a Danish sample in the same study. Furthermore, a recent Chinese study also provided an independent replication of the association of the ALPL locus with kidney stones (PMID: 29489416). Also, in the report by Oddsson et al. a replication of the SLC34A1 and AQP1 loci reported by Urabe et al. was provided (PMID: 22396660). It appears that in total four loci associated with kidney stone disease have been replicated in populations of European and Asian descent.

We thank the reviewer for this comment, now that the paper by Tanikawa et al. has been published further loci have been replicated. We have therefore removed the comment from the text regarding replicated loci, it now states:

Line 64-66 – Main text:

“Four genome-wide association studies of nephrolithiasis have been published identifying fifteen loci associated with disease⁷⁻¹⁰; however no trans-ethnic studies have been undertaken.”

- Page 7-8, line 58-61: For the sake of thoroughness, the number variants tested and significance thresholds used should be clearly stated at this point in the main text.

We thank the reviewer for this suggestion, to clarify the number of variants tested and the significance thresholds the legend for Table 2 has been updated as follows:

Line 695-697 – Table 2:

*“Associations with biochemical phenotypes and number of stone episodes were sought at three loci rs17216707 (CYP24A1), rs838717 (DGKD) and rs1170174 (DGKH, no significant associations were detected, data not shown). *Denotes significance on comparison to bold cohort within group at Bonferroni corrected threshold of $p < 0.05/7 = 0.007$.”*

- Pages 5, lines 79-80: For clarity, the five loci identified to influence CaSR signaling should be clearly specified in the main text.

We thank the reviewer for this suggestion, the gene names have now been added to the text:

Line 177-178 – Main text:

“Five of the identified loci are linked to genes that are predicted to influence CaSR signalling (DGKD, DGKH, WDR72, GIPCI, and BCR) (Fig.2).”

- Supplementary Table 1 and Supplementary Table 2: It would be informative to include the number individuals included in the study for each diagnostic code.

Thank you for this suggestion. We have amended Supplementary Tables, now Tables 5 and 6, accordingly to show the number of individuals with each diagnostic code used in our phenotyping.

Please note that we noticed an omission in our previous manuscript. In addition to ICD-10 and OPCS codes, we used a UK Biobank self-reported diagnostic code for kidney stone surgery to define our cases. Supplementary Table 1 now has an additional column to show the number of individuals with this diagnosis. We have also amended the text in the Online Methods accordingly:

Line 650-652 - Online Methods:

“...data with their medical records^{1,2}. ICD-10 and OPCS codes and a single UK Biobank self-reported operation code for kidney stone surgery were used to identify individuals with a history of nephrolithiasis (Supplementary Table 5).”

1.2 major comments

- Pages 4-5, lines 75-80: Population attributable risk is a measure of the impact of a sequence variant on a disease from a public health point of view. It is however a problematic measurement that is not the right one to evaluate the contribution of sequence variants to disease risk (see paper by Witte et al. for a discussion on the topic PMID: 25223781). For that purpose, another metric such as the sibling recurrent risk ratio might be more appropriate.

As mentioned above, we have now removed the PAR analysis from this paper.

• Page 5, line 96-102:

- To get a clearer picture of the potential recessive effect of the rs17216707 variant in CYP24A1 it would be of interest to:
 - calculate the genotypic odds ratios for kidney stone disease and test if the genotypic effect is consistent with a recessive model.
 - This is best achieved by testing a full genotypic model to estimate the effects of rs17216707[T] heterozygous and homozygous genotypes on kidney stone disease.
 - Significance levels for the heterozygous and homozygous genotypes can be retrieved by comparing the full model to homozygous and heterozygous models, respectively.
 - As this is an important question to address, the analysis would still be of interest if genotypic data is only available for one of the two data sets.
 - It is problematic to conclude on an effect of a genotype when testing traits in a set of cases. If possible, it would be beneficial to also include non stone former controls into the analysis to get a to get a more accurate estimation of the underlying effects. Significance level, effects and confidence intervals for heterozygous and homozygous genotypes should be reported and can be calculated as described above.

We have performed these analyses as suggested by Reviewer 2 using the UK Biobank individual-level data, and have compared the odds ratios for kidney stone disease between the full genotypic model vs a recessive model vs a dominant model. Given that the T allele is the effect allele at this locus, we can derive the following 2×2 tables:

Model			N (cases)	N (controls)
Full genotypic	N (allele)	T	10,898	629,730
	N (allele)	C	2,174	147,286
	Odds ratio (95% CI)	1.17 (1.12-1.23)		
	Z-statistic	6.72		
	P-value	1.71x10 ⁻¹¹		
			N (cases)	N (controls)
Recessive	N (genotype)	TT	4,551	255,261
	N (genotype)	TC + CC	1,985	213,247
	Odds ratio (95% CI)	1.92 (1.82-2.02)		
	Z-statistic	24.0		
	P-value	<2.2x10 ⁻¹⁶		
			N (cases)	N (controls)
Dominant	N (genotype)	TT + TC	6,347	374,469
	N (genotype)	CC	189	14,039
	Odds ratio (95% CI)	1.26 (1.09-1.46)		
	Z-statistic	3.10		
	P-value	0.0019		

This analysis therefore demonstrates that the greatest odds ratio (and z statistic) at this locus is obtained in the recessive model. As suggested by Reviewer 1, we have removed the line in the main text stating that the effect of rs17216707 on serum calcium is “consistent with a recessive effect”, and present instead a p-value for an additive model. However, we do refer to this contingency table analysis in the sentence that follows in the main text, to highlight that the strongest evidence is in favour of a recessive effect of the T allele. We hope that by presenting these data, the reader can form their own interpretation of the likely genotypic model at this locus.

We have commented on this in the Main text, and also include these analyses in the Supplementary Materials:

Line 141-147 – Main Text:

“Individuals homozygous for the CYP24A1 increased-risk allele rs17216707 (T) had a significantly increased mean serum calcium concentration when compared to heterozygotes (mean serum calcium 2.36mmol/l (TT) vs. 2.32mmol/l (TC); trend test p-value for additive effect = 0.023) (Table 2). Analysis of the allelic frequencies of this SNP between cases and controls within the UK Biobank cohort provided strongest evidence in favour of a recessive model (Supplementary Table 4). rs17216707 (T) homozygotes had more kidney....”

Supplementary Materials:

Supplementary Table 4. Comparison of genotypic models for rs1716707. We used individual level data within the UK Biobank dataset to compute the allelic frequencies of the T (effect) and C (non-effect) alleles in renal stone cases and controls, in order to compare the odds ratios in three different genotypic models. The models compared were a full genotypic model (total number of T alleles vs total number of C alleles), a recessive model (TT vs TC+CC), and a dominant model (TT+TC vs CC). The recessive model had the greatest odds ratio and z-statistic.

Model			N (cases)	N (controls)
Full genotypic	N (allele)	T	10,898	629,730
	N (allele)	C	2,174	147,286
	Odds ratio (95% CI)	1.17 (1.12-1.23)		
	Z-statistic	6.72		
	P-value	1.71x10 ⁻¹¹		
			N (cases)	N (controls)
Recessive	N (genotype)	TT	4,551	255,261
	N (genotype)	TC + CC	1,985	213,247
	Odds ratio (95% CI)	1.92 (1.82-2.02)		
	Z-statistic	24.0		
	P-value	<2.2x10 ⁻¹⁶		
			N (cases)	N (controls)
Dominant	N (genotype)	TT + TC	6,347	374,469
	N (genotype)	CC	189	14,039
	Odds ratio (95% CI)	1.26 (1.09-1.46)		
	Z-statistic	3.10		
	P-value	0.0019		

- It is problematic to conclude on an effect of a genotype when testing traits in a set of cases. If possible, it would be beneficial to also include non-stone former controls into the analysis to get a to get a more accurate estimation of the underlying effects. Significance level, effects and confidence intervals for heterozygous and homozygous genotypes should be reported and can be calculated as described above.

The Oxford kidney stone cohort were recruited on the basis of individuals having a diagnosis of kidney stone disease, and no control participants were therefore recruited as part of this study. As such, although we agree with Reviewer 2’s suggestion, we are unfortunately unable to include non-stone former controls into these analyses.

- Page 7-8, line 151-159: To get a clearer picture of the potential difference of effect between the sexes for the rs838717 variant in DGKD it would be of interest to:
- stratify the data by sex calculate the genotypic odds ratios for kidney stone disease and test if the genotypic effects are significantly different between the sexes.

We thank Reviewer 2 for this suggestion. We have performed sex-stratified GWAS of male and female stone formers within the UK Biobank cohort to address this question.

Male GWAS (4,362 cases; 176,832 controls)

Female GWAS (2,174 cases; 211,676 controls)

Association analysis was performed as for the main analysis, conditioning on the same covariates.

At rs838717, we found the following:

Males: OR 1.10 (95% CI 1.06-1.15, $p=8.5 \times 10^{-4}$).

Females: OR 1.11 (95% CI 1.04-1.18, $p=6.2 \times 10^{-6}$).

We obtained the genotypic counts for each genotype (GG / GA / AA) in both males and females and calculated the heterogeneity using MetaGenyo (Martorell-Marugan J et al. BMC Bioinformatics 2017; 18:563). This analysis indicated no evidence of heterogeneity between the two GWAS at this locus:

I^2 statistic: 0; Q-statistic = 0.0013, $p=0.97$.

These findings are therefore consistent with the notion that the failure to detect an association between this locus and urinary calcium excretion in female stone formers was due to a lack of power resulting from the small sample size, as we state in the manuscript.

On balance, we do not feel that this sex-stratified analysis of rs838717 adds a great deal to the interpretation of our results. However, should Reviewer 2 feel that it is prudent to include this analysis in the Supplementary Materials, we would be happy to do so.

- As mentioned above, It is problematic to conclude on an effect of a genotype when testing traits in a set of cases. If possible, it would be beneficial to also include non stone former controls into the analysis to get a to get a more accurate estimation of the underlying effects. Significance level, effects and confidence intervals for heterozygous and homozygous genotypes should be reported and can be calculated as described above

As above, the absence of non-kidney stone former controls in our cohort unfortunately precludes this analysis from being undertaken.

Reviewer #3 (Remarks to the Author):

In this manuscript, the authors reported their findings indicating that genetic test may be needed to identify individuals' risk in kidney stone formation. Authors suggest this as personalized medical care for stone disease towards the precision-medicine approach, by targeting CaSR-signaling or vitamin D activation pathways in patients with recurrent kidney stone former. The topic itself is valuable and the method based on genome-wide association studies and meta-analysis, seems useful for the treatment of a subset of patients with nephrolithiasis. However, there are some fundamental aspects of CaSR (GpCR) activation i.e. receptor desensitization and adaptation were not discussed.

We thank the reviewer for these supportive comments and observations regarding our studies of the CaSR-signaling pathway. In this study we have undertaken *in vitro* work with the aim of demonstrating that alterations in the

expression level of DGKD affect CaSR-signaling. Whilst we appreciate that there are many more aspects of CaSR-signaling that could be considered, desensitization is only thought to have a minimal impact on CaSR function, in contrast to other GPCRs (Conigrave and Ward, Best Pract Res Clin Endocrinol Metab 2013, PMID: 23856262), and was therefore not pursued in these studies.

Moreover, in Figure 2 they did not mention about the CaSR signaling mediated activation Ca²⁺ entry pathway, perhaps by TRPC/SOCE; so, any dynamic Ca²⁺ mobilization pathway(s) was not considered in in-vitro functional experiments.

We thank the reviewer for these comments and in light of this observation we have now undertaken *in vitro* studies of the role of DGKD in CaSR-signaling via intracellular calcium mobilization. Figure 2 notes the role of intracellular calcium release in the CaSR-signaling pathway in the box “CaSR signaling pathways including: [Ca²⁺]_i, MAPK, cAMP decrease”. These studies of intracellular calcium mobilization have revealed that this signaling pathway is unaffected by DGKD knockdown, and therefore may provide an explanation for the observed correlation of DGKD increased-risk allele rs838717 (G) with increased urinary calcium excretion but not serum calcium concentration.

Line 280-301– Main Text:

“Furthermore, to investigate the role of DGKD in CaSR-signaling via intracellular calcium mobilization, HEK-CaSR-NFAT and HEK-CaSR cells were treated with scrambled or DGKD targeted siRNA and intracellular NFAT and intracellular calcium responses to alterations in extracellular calcium concentration assessed via NFAT-reporter and Fluo-4 calcium assays, respectively. Intracellular calcium responses were unaffected by a reduction in DGKD expression (NFAT maximal response DGKD-KD=2.91 fold change, 95% CI=2.69-3.12 vs. WT=2.85 fold change, 95% CI=2.62-3.08, p=0.73; Fluo-4 maximal response DGKD-KD=96.16, 95% CI=89.83-102.5 vs. WT=96.78 fold change, 95% CI=92.76-100.8, p=0.83) (Supplementary Fig. 5D and E).

These findings provide evidence that DGKD influences CaSR-mediated signal transduction and suggest that the DGKD increased-risk allele may associate with a relative increase in DGKD expression thereby enhancing CaSR-mediated signaling via the MAPK pathway whilst leaving signaling via the intracellular calcium pathway unaffected. This biased signaling may provide an explanation for the observed correlation of the DGKD increased-risk allele rs838717 (G) with increased urinary calcium excretion but not serum calcium concentration (Table 2).....We predict that the development of biased calcilytics may provide a novel, targeted therapeutic approach to reduce urinary calcium excretion in recurrent stone formers carrying CaSR-associated increased-risk allele.”

Supplementary Figure 5

Figure 5. CaSR-mediated responses following DGKD knockdown in HEK-CaSR and HEK-CaSR-NFAT cells. Panel A shows relative expression of *CASR*, as assessed by quantitative real-time PCR of HEK-CaSR-SRE cells treated with scrambled (WT) or *DGKD* (DGKD-KD) siRNA. Samples were normalized to a geometric mean of four housekeeper genes: *PGK1*, *GAPDH*, *TUB1A*, *CDNK1B*. n=8. Panel B shows the relative expression of CaSR, as assessed by densitometry of western blots from cells treated with scrambled or *DGKD* siRNA. Samples were normalized to *PGK1*. n=6. Panel C shows pERK responses of HEK-CaSR cells in response to changes in extracellular calcium concentration. Cells were treated with scrambled (WT) or *DGKD* (DGKD-KD) siRNA. The responses \pm SEM are shown for 4 independent transfections for WT and DGKD-KD cells. Treatment with *DGKD* siRNA led to a reduction in maximal response (red line) compared to cells treated with scrambled siRNA (black line). Panel D shows NFAT responses of HEK-CaSR-NFAT cells in response to changes in extracellular calcium concentration. Cells were treated with scrambled (WT) or *DGKD* (DGKD-KD) siRNA. The responses \pm SEM are shown for 5 independent transfections for WT and DGKD-KD cells. Treatment with *DGKD* siRNA did not affect the maximal response (red line) compared to cells treated with scrambled siRNA (black line). Post desensitization points were not included in the analysis (grey, and light red). Panel E shows intracellular calcium responses of HEK-CaSR cells in response to changes in extracellular calcium concentration. Cells were treated with scrambled (WT) or *DGKD* (DGKD-KD) siRNA. The responses \pm SEM are shown for 4 independent transfections for WT and DGKD-KD cells. Treatment with *DGKD* siRNA did not affect the maximal response (red line) compared to cells treated with scrambled siRNA (black line). Statistical comparisons of maximal response were undertaken using F test. Students T tests were used to compare relative expression. Two-way ANOVA was used to compare points on dose response curve with reference to WT. Data are shown as mean \pm SEM with *p<0.05, **p<0.01, ****p<0.0001.

Although, in a similar study, has already been reported by the authors (Cell Reports 22, 1054–1066), that CaSR mediated sustained signaling can occur by a non-canonical endosomal pathway, which they provide an explanation for the observed reduction in CaSR signaling because of AP2s mutation. CaSR have been shown as a part of Ca²⁺ channel signaling complex, expression of which increased at the plasma membrane due to CaSR agonist activation.

The *in vitro* studies undertaken in this manuscript focus only on knockdown of DGKD, a protein which is not predicted to effect CaSR endocytosis (Figure 2) and therefore would not be expected to directly alter signaling via the non-canonical pathway. As noted in Figure 2, WDR72 and GIPC1 may indeed effect non-canonical signalling, however we felt that studies of these proteins was beyond the scope of this manuscript; we hope that these investigations will form the basis of future studies.

Most importantly, the authors presented the experimental results, but did not attempt to clarify why the effect of CaSR can be diverse in different conditions. The mechanism behind the experimental results may be much more interesting.

As noted above, we thank the reviewer for prompting us to undertake studies of the effect of DGKD knockdown on CaSR-signaling via alterations in intracellular calcium concentration. As the reviewer predicted these findings provide additional interest and may shed light on the phenotype observed in carriers of the DGKD increased-risk allele rs838717 (G):

Line 290-301– Main Text:

“These findings provide evidence that DGKD influences CaSR-mediated signal transduction and suggest that the DGKD increased-risk allele may associate with a relative increase in DGKD expression thereby enhancing CaSR-mediated signaling via the MAPK pathway whilst leaving signaling via the intracellular calcium pathway unaffected. This biased signaling may provide an explanation for the observed correlation of the DGKD increased-risk allele rs838717 (G) with increased urinary calcium excretion but not serum calcium concentration (Table 2). Calcilytics, including NPS-2143 and ronacaleret, rectify enhanced CaSR-mediated signaling in vitro and biochemical phenotypes in mouse models of ADH³⁰⁻³². We predict that the development of biased calcilytics may provide a novel, targeted therapeutic approach to reduce urinary calcium excretion in recurrent stone formers carrying CaSR-associated increased-risk alleles.”

Some more comments need to be addressed:

1. Authors did some *in vitro* expression study, but did not try to show how much influence of CaSR mediated signaling canonical vs non-canonical pathway.

We thank the reviewer for this comment, as noted above, the *in vitro* studies undertaken in this manuscript focus only on knockdown of DGKD, a protein which is not predicted to effect CaSR endocytosis (Figure 2) and therefore would not be expected to directly alter signalling via non-canonical pathway. As noted in Figure 2, WDR72 and GIPC1 may indeed effect non-canonical signalling, however we felt that studies of these proteins was beyond the scope of this manuscript; we hope that these investigations will form the basis of future studies.

2. It is not clear why the authors used HEK 293-stably expressed, stable expression sometime goes up and down depending on the condition, there are inducible system used for such purposes. Some study shows that CaSR in HEK293 cells stably expressing human CaSR (HEK-CaSR cells) varies when glucose concentration in the buffer was raised as glucose activate CSR.

We thank the reviewer for these comments. To address these concerns that CaSR-expression may not be comparable between experimental groups we have undertaken quantitation using both qRT-PCR and western blot analysis in combination with densitometry, see Supplementary Figure 5 above. We have previously investigated the effects of glucose on CaSR activation and found that alterations in glucose concentrations from 3 to 25 mM did not affect CaSR-signaling responses in HEK-CaSR cells (Babinsky et al. Endocrinology 2017, PMID:28575322). Furthermore, the glucose concentration in the media of the cells studied in this manuscript was constant throughout the assays undertaken.

3. Over-expression of CSR then siRNA? Native CSR may be good, endogenous method to show regulation can show a comparative situation in native cells.

We thank the reviewer for this comment, HEK-293 cells do not endogenously express the CaSR and hence we used these well-established HEK-CaSR cells (Nesbit et al. Nat Gen. 2013, PMID: 23222959) as a basis for our *in vitro* work. Rather than using siRNA to target the CaSR, siRNA was used to reduce expression of DGKD. Our studies have provided evidence that DGKD expression influences CaSR-signaling via the MAPK pathway.

4. CaSR western blot presented in Fig3B is questionable with multiple bands (no control), which needs to be quantified, or RT-PCR to show knockdown (with full gel). Moreover, this is not definitive test, they should add staining and/or gene expression data with WB and/or qRT-PCR to show CaSR and DGKD expression in WT and DGKD-KD.

We thank the reviewer for this comment, we have improved the clarity of the western blot presented in Figure 3 as shown below. In addition, we have quantified the expression of the DGKD and CaSR via qRT-PCR and western blot in Figure 3 and Supplementary Figure 5. This is also noted in the main text:

Line 259-261– Main Text:

“Treatment with DGKD targeted siRNA resulted in a reduction in DGKD expression when compared to cells treated with scrambled siRNA without alteration in CaSR expression (Fig. 3A-C and Supplementary Fig. 5A and B).”

Figure 3. CaSR-mediated SRE responses following DGKD knockdown and effect of cinacalcet treatment in HEK-CaSR-SRE cells. Panel A shows relative expression of *DGKD*, as assessed by quantitative real-time PCR of HEK-CaSR-SRE cells treated with scrambled (WT) or *DGKD* (DGKD-KD) siRNA and used for SRE experiments. Samples were normalized to a geometric mean of four housekeeper genes: *PGK1*, *GAPDH*, *TUB1A*, *CDNK1B*. n=8. Panel B shows a representative western blot of lysates from HEK-CaSR cells treated with scrambled or *DGKD* siRNA and used for SRE experiments. PGK1 was used as a loading control. Panel C shows the relative expression of *DGKD*, as assessed by densitometry of western blots from cells treated with scrambled or *DGKD* siRNA demonstrating a ~50% reduction in expression of *DGKD* following treatment with *DGKD* siRNA. Samples were normalized to PGK1. n=6. Panel D shows SRE responses of HEK-CaSR-SRE cells in response to changes in extracellular calcium concentration. Cells were treated with scrambled (WT) or *DGKD* (DGKD-KD) siRNA. The responses \pm SEM are shown for 8 independent transfections for WT and DGKD-KD cells and 4 independent transfections for DGKD-KD + 5nM cinacalcet cells. Treatment with *DGKD* siRNA led to a reduction in maximal response (red line) compared to cells treated with scrambled siRNA (black line). This loss-of-function could be rectified by treatment with 5nM cinacalcet (blue line). Post desensitization points were not included in the analysis (grey, light red, and light blue). Panel E demonstrates the mean maximal responses with SEM of cells treated with scrambled siRNA (WT, black), *DGKD* siRNA (DGKD-KD, red) and *DGKD* siRNA incubated with 5nM cinacalcet (blue). Statistical comparisons of maximal response were undertaken using F test. Students T tests were used to compare relative expression. Two-way ANOVA was used to compare points on dose response curve with reference to WT. Data are shown as mean \pm SEM with **p<0.01, ****p<0.0001.

5. Additional evidence is required to justify the assessment of intracellular MAPK responses. A suggestion would be to evaluate ERK phosphorylation in WT and DGKD-KD to assess downstream MAPK responses. Other downstream MAPK responses would also be appropriate here. Which is also important for evaluating downstream CaSR signaling?

We thank the reviewer for this comment. In light of this suggestion we have confirmed our finding that knockdown of *DGKD* affects CaSR-signaling via the MAPK signalling pathway by assessing ERK phosphorylation in WT and DGKD-KD cells. These findings are detailed in the main text and presented in Supplementary Figure 5 as shown above.

Line 255-267 – Main Text:

“To investigate the role of DGKD in CaSR-signalling via the MAPK pathway, HEK-CaSR-SRE and HEK-CaSR cells were treated with scrambled or DGKD targeted siRNA and intracellular MAPK responses to alterations in extracellular calcium concentration assessed via SRE and ERK-phosphorylation (pERK) assays, respectively. Treatment with DGKD targeted siRNA resulted in a reduction in DGKD expression when compared to cells treated with scrambled siRNA without alteration in CaSR expression (Fig. 3A-C and Supplementary Fig. 5A and B). SRE and pERK responses were significantly decreased in cells with reduced DGKD expression (DGKD-KD) when compared to cells with baseline DGKD expression (WT) (SRE maximal response DGKD-KD=5.28 fold change, 95% confidence interval (CI)=4.77-5.79 vs. WT=7.20 fold change, 95% CI=6.46-7.93, p=0.0065; pERK maximal response DGKD-KD=24.77, 95% CI=22.16-27.38 vs. WT=39.46 fold change, 95% CI=34.07-44.84, p=0.0056).”

Minor comments:

1. DMSO concs. were not mentioned as DMSO can also increase [Ca²⁺]_i release.

We thank the reviewer for this observation, the DMSO concentration was consistent across assays at 0.0001%. This has been noted in the online methods:

Line 914-917 – Online Methods:

“At 72 hours the media was changed to varying concentrations of extracellular calcium (0.1-5mM), with either 5nM cinacalcet or equivalent volume of DMSO (final concentration of DMSO 0.0001%), and the cells were incubated for a further 4 hours at 37°C.”

2. What is the rationale for using 0.05% FBS media with 0.45mM calcium for 12 hours.

Thank you for raising this question, we have now clarified the rationale for using this media in the online methods:

Line 911-914 – Online Methods:

“At 60 hours post transfection, HEK-CaSR-SRE cells were incubated in 0.05% fetal bovine serum media with 0.45mM calcium for 12 hours, reducing extracellular calcium concentration and thus inducing basal cellular CaSR-mediated responses whilst maintaining cellular viability.”

3. In Results the figures referred as Fig. 3A-E, make sure that text and figures are matched as appropriate.

We thank the reviewer for spotting this error, the text has now been modified:

Line 259-261 – Main Text:

“Treatment with DGKD targeted siRNA resulted in a reduction in DGKD expression when compared to cells treated with scrambled siRNA without alteration in CaSR expression (Fig. 3A-C and Supplementary Fig. 5A and B).”

Line 267-278 – Main Text:

“Cinacalcet rectified this loss-of-function in SRE-reporter assays (DGKD-KD+5nM cinacalcet, maximal response=7.62 fold change, 95% CI=5.98-9.27)(Fig. 3D-E and Supplementary Fig.5C).”

Editors comments

Please be aware that for certain types of new data, including most types of genetic data, journal policy is that deposition in a community-endorsed, public repository is generally mandatory prior to publication. Data submission can be a lengthy process, and we strongly suggest that you begin this well in advance of potential publication to avoid delays later on. Please include a statement about data availability in your point-by-point letter accompanying your revisions.

Thank you for this note and request. We added the following data availability statement to the manuscript

Line 977-981 – Online Methods:

“Full UK Biobank data are available by the direct application to UK Biobank. Full GWAS summary statistics from the UK Biobank GWAS and the UK-Japanese meta-analysis can be found at: doi TBC. Source data for Figure 3, Table 2, and Supplementary Figure 5 are provided with the paper. All other relevant data is available from the authors on request.”

The process of uploading the meta-analysis summary statistics to *Research Data Oxford* is not time consuming but the data cannot be altered once a doi has been assigned. We will therefore upload the summary statistics to *Research Data Oxford* if the manuscript is accepted for publication and modify this statement accordingly.

Reviewers' Comments:

Reviewer #1:

Remarks to the Author:

The authors had addressed several of my concerns but there is still few issues to be resolved. Of the 7 novel loci described, 4 only reached genome-wide significance when combining discovery and replication, so these are not really replicated. The BCR locus did not replicate in the JBB, likely due to the low frequency of the variant in Japanese. These findings should be better highlighted in the paper, given the authors focus on trans-ethnic analyses.

Abstract conclusion (lines 39 to 42) and main text (lines 116-128) related to recommendations for genotyping of variants to guide therapy and inform risk are premature given the low effect size of the variant (OR 1.19). These conclusions need to be revised.

The identified potential phenotypic heterogeneity between UKB and JBB should be discussed as a limitation of the research. This will not diminish the findings of this study but can be helpful for future research in the field. Additional limitations to be discussed are the inclusion of participants taking medications that can affect the risk of kidney stone (steroids increasing and diuretics being protective), and the case-only design for the Oxford cohort.

The findings of an association of the CYP24A1 variant with number of kidney stones episodes is a excellent use of a case-only study and could be better highlighted in the abstract.

I don't think reporting the allele differences between cases and controls is what the reviewers suggested.

Is there any evidence for variants being expression quantitative loci for a gene?

The paragraph related to CYP24A1 variant genotyping for treatment purpose is premature given the low effect size of the variant (19% increase risk) and the need to additional work because

Figure 1. The description of UKB participants as patients is likely not appropriate as they were recruited into the study without selection for having a clinical disease.

The new Figure S1 regional plots based on LD from European and East Asian ancestry: please comments if there anything learned about using these two population for gene discovery besides the additional of samples.

Table S2. Three variants show large heterogeneity in effects between UKB and JBB, please add the heterogeneity p-values in the footnote of main Table 1.

Reviewer #2:

Remarks to the Author:

Thank you for your response.

I would like to point out that in Table 1 the SCNN1B locus is not marked as novel (with an asterix) as it has been in figure 1.

Regarding potential difference of effect between the sexes for the rs838717 variant in DGKD. There is no need to include the sex-stratified analysis in the manuscript.

Reviewer #3:

Remarks to the Author:

The authors use pharmacological approach, combined with fluorescence calcium measurement to collect the functional data for CaSR to show urinary/serum biomarkers of calcium metabolism, and performed in vitro studies of cells with reduced DGKD expression. Main purpose is that understanding the genetic susceptibility to kidney stones, which will help to identify potential

therapies for kidney stone disease. It is good to see that the Authors have addressed the concerns from the previous version.

Still there are a few minor points that still came to my attention, which I would like to see addressed before publication. These are listed below. Other than that, I think the manuscript has greatly improved and I recommend accepting it for publication.

1. Figure 3B CaSR Blot looks darker background, with 2 bands. Authors need to be marked (}/*) which band they are considering. They should submit a full blot in supplementary. There is no loading control for this western blot, which is necessary.
2. In Fluo-4 measurement of Ca²⁺ concentration did they use EGTA in addition to Ionomycin? Authors need to add the details of the data analysis of Ca²⁺ measurement.
3. Similarly, in Supplementary Figure 5, "CaSR-mediated responses following DGKD knockdown in HEK-CaSR and HEK-CaSR-NFAT cells" – Details need to add in "Panel E shows intracellular calcium responses of HEKCaSR cells in response to changes in extracellular calcium concentration."
4. Concentrations of scrambled (WT) and DGKD (DGKD-KD) siRNA also needs to be added.
5. Figure 3D label inside the graph looks odd since nothing indicated in it.

Resubmission of: Genetic variants of calcium and vitamin D metabolism in kidney stone disease by Howles et al.

Reviewer #1 (Remarks to the Author):

The authors had addressed several of my concerns but there is still few issues to be resolved. Of the 7 novel loci described, 4 only reached genome-wide significance when combining discovery and replication, so these are not really replicated. The BCR locus did not replicate in the JBB, likely due to the low frequency of the variant in Japanese. These findings should be better highlighted in the paper, given the authors focus on trans-ethnic analyses.

Of the seven novel loci, we can divide them into three groups based on association p-values (loci shown in parentheses):

- 1) Genome-wide significant in UKB and nominally significant in JBB (*SLC22A2, HIBADH*).
- 2) Nominally significant in both UKB and JBB; genome-wide significant on meta-analysis (*DGKD, ABCG2, SCNN1B, SOX9*).
- 3) Genome-wide significant in UKB and non-significant in JBB (*BCR*).

With regard to group 2, the Reviewer highlights that these loci only achieved genome-wide significance on meta-analysis and that they were not really replicated. We respectfully point out that we have not stated anywhere in the manuscript that these seven novel loci *replicated*; we have merely stated that they are significant on meta-analysis. We hope the Reviewer agrees with us that the possibility of discovering genome-wide significant loci that do not reach the required threshold in a single population is the very reason for performing a meta-analysis.

However, we do agree with the reviewer that we should better highlight trans-ethnic differences in the *BCR* locus in our manuscript, so we have done so in two places (underlined):

Line 97-104 – Main text

“The allelic effects were concordant across both studies at all 20 loci, with minimal evidence of heterogeneity between the two GWAS at the majority of loci, with a Q-statistic p-value > 0.05 at 17/20 loci (Supplementary Table 2), suggesting that the genetic architecture of kidney stone disease is very similar between populations of European and East Asian ancestry. For the seven novel loci, we found that the effects are directionally concordant in both datasets and of the same magnitude, with the exception of rs13054904 (the BCR locus). This locus therefore highlights ethnic differences in the predisposition to renal stone disease”.

Line 163-169 – Main text

“rs13054904 is ~110kb upstream of BCR, encoding a RAC1 (Rac Family Small GTPase 1) GTPase-activating protein known as Breakpoint Cluster Region (BCR). Of the 20 genome-wide significant loci discovered in the meta-analysis of British and Japanese populations, this was the only locus that did not reach nominal significance in one of the two populations (British: $p = 3.30 \times 10^{-12}$; Japanese: $p = 0.505$), suggesting that this locus predisposes to renal stones in European but not East Asian populations.”

Abstract conclusion (lines 39 to 42) and main text (lines 116-128) related to recommendations for genotyping of variants to guide therapy and inform risk are premature given the low effect size of the variant (OR 1.19). These conclusions need to be revised.

We thank the reviewer for this comment, it is interesting to note that effect size of a GWAS variant dictating its clinical utility has little relation to the biological effect of a drug targeting a biological pathway, as exemplified by *CTLA4* in rheumatoid arthritis:

A meta-analysis of rs231775, a variant in the *CTLA4* gene found the odds ratio of its association with rheumatoid arthritis to be 1.18 (95% CI 1.04-1.34) (Li et al. *J Clin Immunol*, 2012). Despite this modest odds ratio (comparable to what we found for rs17216707), this molecule is being used successfully as a therapeutic target in the form of Abatecept, a T-cell co-stimulation modulator that consists of the extracellular domain of human CTLA-4 (Guo et al., *Bone Research*, 2018). This example neatly illustrates that the effect size in GWAS does not correspond to biological effects in treatment.

However, we do accept that further studies are required and to make this clear we to make it clear that further studies are required we have altered the text as follows:

Line 38-41 – Abstract

“Our findings indicate that studies of genotype-guided precision-medicine approaches, including withholding vitamin D supplementation and targeting vitamin D activation or CaSR-signaling pathways in patients with recurrent kidney stones, are warranted.”

Line 252-254 – Main text

“However, our findings suggest that supplementation may put individuals homozygous for the CYP24A1 increased-risk allele at risk of kidney stones; studies are required to investigate this hypothesis further.”

Line 285-290 – Main text

“Our findings suggest that there may be a role for genetic testing to identify individuals in whom vitamin D supplementation should be used with caution and to facilitate a precision-medicine approach for the treatment of recurrent kidney stone disease, whereby targeting of the CaSR-signaling or vitamin D metabolism pathways may be beneficial in the treatment of a subset of patients with nephrolithiasis; further studies are required.”

The identified potential phenotypic heterogeneity between UKB and JBB should be discussed as a limitation of the research. This will not diminish the findings of this study but can be helpful for future research in the field.

We thank the reviewer for this comment, we have now highlighted the potential phenotypic heterogeneity between the two cohorts:

Line 78-83 – Main text

“Stone forming individuals were identified from the UK Biobank using ICD-10 and OPCS codes and a single UK Biobank self-reported operation code for kidney stone surgery and excluded if recorded to have a disorder known to predispose to kidney stone disease (Supplementary Table 1 and 2). In contrast, Biobank Japan cases were diagnosed and enrolled by physicians; information regarding conditions known to predispose to kidney stones was unavailable.”

Additional limitations to be discussed are the inclusion of participants taking medications that can affect the risk of kidney stone (steroids increasing and diuretics being protective), and the case-only design for the Oxford cohort. The findings of an association of the CYP24A1 variant with number of kidney stones episodes is an excellent use of a case-only study and could be better highlighted in the abstract.

Thank you for this comment, we have highlighted the case-only design in the main text and abstract, along with the inclusion of patients taking medications in the main text:

Line 34-36 - Abstract

“In a validation cohort of only nephrolithiasis patients, the CYP24A1-associated locus correlates with serum calcium concentration and number of nephrolithiasis episodes and the DGKD-associated locus correlates with urinary calcium excretion.”

Line 130-132 – Main text

“Only stone forming patients were recruited, 15% of individuals were taking medications, including steroids and diuretics, that may affect risk of kidney stone formation.”

I don't think reporting the allele differences between cases and controls is what the reviewers suggested.

With regard to the reporting of allelic differences between cases and controls, we have addressed this point in full in response to Reviewer 2's comments, below.

Is there any evidence for variants being expression quantitative loci for a gene?

With regard to expression quantitative trait loci (eQTL), we have scrutinised the index SNPs at our 20 genome-wide significant in the GTEx database, and found that 15 of the SNPs are eQTL in a variety of different tissues. While we are happy to present these data as a Supplementary Data file if the Reviewer or Editor feel it is valuable to do so, we question the relevance of this given the importance of context- and tissue-specificity of eQTL.

The most informative piece of data in this regard, we feel, is the MAGMA tissue expression analysis of our GWAS-summary data (Supplementary Figure 2, showing the relationships between highly expressed genes in a specific tissue and the GWAS loci), which demonstrate that by far the most significantly enriched tissue is “kidney cortex”.

The paragraph related to CYP24A1 variant genotyping for treatment purpose is premature given the low effect size of the variant (19% increase risk) and the need to additional work because

Thank you for this comment, please see above for our note regarding effect size of a GWAS variant dictating its clinical utility in relation to the biological effect of a drug targeting a biological pathway, as exemplified by *CTLA4* in rheumatoid arthritis. However, in light of this concern we have altered the text as follows:

Line 38-41 – Abstract

“Our findings indicate that studies of genotype-guided precision-medicine approaches, including withholding vitamin D supplementation and targeting vitamin D activation or CaSR-signaling pathways in patients with recurrent kidney stones, are warranted.”

Line 252-254 – Main text

“However, our findings suggest that supplementation may put individuals homozygous for the CYP24A1 increased-risk allele at risk of kidney stones; studies are required to investigate this hypothesis further.”

Line 285-290 – Main text

“Our findings suggest that there may be a role for genetic testing to identify individuals in whom vitamin D supplementation should be used with caution and to facilitate a precision-medicine approach for the treatment of recurrent kidney stone disease, whereby targeting of the CaSR-signaling or vitamin D metabolism pathways may be beneficial in the treatment of a subset of patients with nephrolithiasis; further studies are required.”

Figure 1. The description of UKB participants as patients is likely not appropriate as they were recruited into the study without selection for having a clinical disease.

Thank you for this observation, the figure legend text has been altered:

Line 730-731 – Main text

“A trans-ethnic meta-analysis of kidney stone disease was performed for 12,123 individuals with kidney stone disease and 416,928 controls from the UK Biobank and BioBank Japan.”

The new Figure S1 regional plots based on LD from European and East Asian ancestry: please comments if there anything learned about using these two population for gene discovery besides the additional of samples.

One might conceivably use the different LD blocks in the Japanese population vs. European population to narrow down the credible set of candidate causal SNPs at each locus. However, further analyses are needed to formally use this information in a Bayesian framework, and we believe that this is beyond the scope of this paper.

Table S2. Three variants show large heterogeneity in effects between UKB and JBB, please add the heterogeneity p-values in the footnote of main Table 1.

Thank you for this suggestion, the footnote of Table 1 has been altered as follows:

Line 784-785 – Main text

“[†]These three variants showed nominally significant heterogeneity in effects between the two populations (Q value $p < 0.05$).”

Reviewer #2 (Remarks to the Author):

Thank you for your response.

I would like to point out that in Table 1 the SCNN1B locus is not marked as novel (with an asterisk) as it has been in figure 1.

Thank you for this comment, please see Table 1 below, the SCNN1B locus is marked with an asterisk:

Chr.	SNP					Discovery GWAS in UK Biobank				Replication GWAS in BioBank Japan				Meta-Analysis		Candidate gene
	SNP	Position	Annotation	EA ¹	NEA ¹	EAF ²	INFO ³	OR ⁴	P ⁵	EAF ²	INFO ³	OR ⁴	P ⁵	OR ⁶	P ⁷	
1	rs10917002	21836340	I	T	C	0.11	0.997	1.18 (1.12-1.25)	3.60×10⁻⁸	0.38	0.998	1.09 (1.04-1.15)	5.83×10 ⁻¹	1.13 (1.09-1.17)	3.45×10⁻¹¹	ALPL ¹
2	rs780093	27742603	I	T	C	0.38	1	1.08 (1.04-1.12)	3.60×10 ⁻¹	0.56	0.997	1.14 (1.09-1.18)	1.10×10⁻⁸	1.10 (1.08-1.13)	1.31×10⁻¹¹	GCKR
2	rs13003198*	234257105	IG	T	C	0.39	0.997	1.10 (1.06-1.14)	6.50×10 ⁻³	0.25	0.98	1.12 (1.06-1.18)	1.09×10 ⁻¹	1.11 (1.07-1.14)	3.89×10⁻¹¹	DGKD
4	rs1481012*	89039082	I	G	A	0.11	0.994	1.12 (1.06-1.18)	4.30×10 ⁻¹	0.30	0.994	1.11 (1.05-1.17)	1.50×10 ⁻¹	1.11 (1.07-1.16)	2.79×10⁻⁸	ABCG2
5	rs56235845	176798040	S	G	T	0.33	0.986	1.16 (1.12-1.20)	9.10×10⁻¹¹	0.31	0.87	1.18 (1.12-1.25)	1.88×10⁻¹¹	1.16 (1.13-1.20)	2.64×10⁻¹¹	SLC34A1
6	rs1155347	39146230	IG	C	T	0.22	0.975	1.12 (1.07-1.17)	2.60×10 ⁻¹	0.16	0.925	1.16 (1.08-1.24)	1.33×10 ⁻⁶	1.13 (1.09-1.17)	8.54×10⁻¹¹	KCNK5
6	rs77648599*	160624115	I	G	T	0.03	0.992	1.33 (1.21-1.47)	5.50×10⁻⁹	0.04	0.739	1.22 (1.06-1.44)	1.89×10 ⁻¹	1.30 (1.20-1.42)	5.39×10⁻¹¹	SLC22A2
7	rs12539707*	27626165	I	T	C	0.30	0.999	1.13 (1.08-1.17)	6.30×10⁻¹⁰	0.09	0.789	1.10 (1.01-1.21)	0.0268	1.12 (1.08-1.16)	1.09×10⁻¹⁰	HIBADH
7	rs12666466	30916430	I	G	C	0.03	0.994	1.22 (1.11-1.34)	5.00×10 ⁻¹	0.12	0.989	1.17 (1.08-1.26)	2.80×10 ⁻⁶	1.19 (1.12-1.26)	3.26×10⁻⁸	AQP1
11	rs4529910	111243102	I	T	G	0.27	0.998	1.07 (1.02-1.11)	1.40×10 ⁻¹	0.59	0.999	1.12 (1.08-1.16)	3.94×10 ⁻⁷	1.09 (1.06-1.12)	4.25×10⁻¹⁰	POU2AF
13	rs1037271	42779410	I	C	T	0.39	0.995	1.11 (1.07-1.15)	2.50×10⁻⁸	0.55	0.936	1.20 (1.15-1.24)	7.49×10⁻¹¹	1.15 (1.12-1.18)	1.29×10⁻¹¹	DGKIF ¹
15	rs578595	53997089	I	C	A	0.46	0.996	1.09 (1.05-1.13)	2.50×10 ⁻⁶	0.69	0.996	1.11 (1.06-1.15)	2.25×10 ⁻¹	1.09 (1.07-1.12)	6.26×10⁻¹¹	WDR72
16	rs77924615	20392332	I	A	G	0.20	0.980	1.13 (1.08-1.18)	1.80×10⁻⁸	0.22	0.984	1.17 (1.10-1.24)	2.80×10⁻⁸	1.14 (1.10-1.19)	1.14×10⁻¹¹	UMOD
16	rs889299*	23381914	I	G	A	0.76	1	1.10 (1.05-1.14)	8.20×10⁻¹⁰	0.66	0.895	1.09 (1.04-1.14)	9.39×10⁻¹¹	1.09 (1.06-1.13)	1.55×10⁻¹¹	SCNN1B
17	rs1010269	59448945	I	G	A	0.83	0.981	1.08 (1.03-1.14)	7.10×10 ⁻¹	0.56	0.87	1.17 (1.12-1.22)	4.82×10⁻¹¹	1.13 (1.10-1.17)	3.71×10⁻¹¹	BCAS1
17	rs4793434*	70352537	I	G	C	0.50	0.993	1.09 (1.05-1.13)	1.50×10 ⁻⁶	0.32	0.983	1.09 (1.04-1.15)	2.04×10 ⁻¹	1.09 (1.06-1.12)	4.52×10⁻¹⁰	SOX9
19	rs3760702	14588237	IG	A	G	0.33	0.994	1.08 (1.05-1.13)	1.40×10 ⁻¹	0.25	0.971	1.14 (1.08-1.20)	3.78×10 ⁻¹	1.09 (1.07-1.13)	1.98×10⁻¹⁰	GIPC1
20	rs17216707	52732362	IG	T	C	0.81	0.961	1.17 (1.12-1.22)	9.90×10⁻¹¹	0.92	0.766	1.24 (1.15-1.34)	5.90×10 ⁻⁶	1.19 (1.14-1.23)	7.82×10⁻¹⁰	CYP24A1
21	rs12626330	37835982	I	G	C	0.49	0.980	1.16 (1.12-1.20)	5.80×10⁻¹¹	0.39	0.981	1.12 (1.07-1.18)	2.77×10 ⁻¹	1.15 (1.12-1.18)	7.24×10⁻¹¹	CLDN14
22	rs13054904*	23410918	I	A	T	0.26	0.999	1.15 (1.11-1.20)	3.30×10⁻¹¹	0.02	0.967	1.05 (0.91-1.26)	0.505	1.14 (1.10-1.19)	4.49×10⁻¹¹	BCR

Regarding potential difference of effect between the sexes for the rs838717 variant in DGKD. There is no need to include the sex-stratified analysis in the manuscript.

Thank you.

Regarding the analysis of the potential recessive effects of the rs17216707 variant in CYP24A1 (Page 5, line 96-102)

The full genotype model should include parameters for both heterozygotes and homozygotes. In such a model the SNP is coded by two indicator variables, one for heterozygous carrier of the minor allele (X1) and one for homozygous carrier of the minor allele (X2). In other words, we convert AA into “X1=0, X2=0”, AB into “X1=1, X2=0”, and BB into “X1=0, X2=1” where A stands for the common allele and B for the minor allele. The response variable Y is coded as 1 for cases and 0 for controls.

$$\text{Log}(P(Y = 1|X1, X2) / (1 - P(Y = 1|X1, X2))) = \beta_0 + \beta_1X1 + \beta_2X2.$$

This model, which gives effects for both heterozygotes and homozygotes, can then be compared (via statistical model comparison) to other nested models, e.g. the recessive model to determine if there is a significant effect for heterozygotes only. If not the authors can conclude that the homozygotes are driving the association and the recessive model is the best one. This is still not clear from the table added after the first revision.

For our analyses of the full genotypic, dominant and recessive models for rs17216707, we used well-established definitions of these three models, as outlined in: Clark et al., “Basic statistical analysis in genetic case-control studies”, *Nat Protoc*, 2011 (specifically, see paragraph 2 of the section titled “Tests of Association”). However, we performed the analyses requested by the reviewer, although there were certain limitations to these, as outlined below.

We modelled our data according to the reviewer’s instructions using a combination of linear predictors (X1 X2) to include heterozygous (X1 = 1, X2 = 0) and homozygous for the **minor** allele (X1 = 0, X2 = 1) in the fitting (**please note that the major allele is associated with renal stones**). Using the likelihood ratio test with single term deletion, we found that the model was significantly more different to the full genotype model when X1 was removed (p = 5.704 x 10⁻¹⁰ v p = 7.946 x 10⁻⁵), and concluded that the effect of association between rs17216707 and renal stones is mainly driven by the heterozygous genotype rather than homozygous for the low-risk allele.

There isn’t a known way to create a nested model that explains both the recessive and dominant model as they are coded differently, where in the case of rs17216707, recessive was coded as TT = 0, TC = 1 and CC = 1, and dominant was TT = 0, TC = 0 and CC = 1. And since the two models can’t be nested, they can’t be compared. We can however determine the best fit model by calculating each model’s respective AIC. Using the coding scheme for recessive and dominant model described above and logistic regression, the AIC of rs17216707 for the full genotypic model (0, 1 and 2 representing each of the genotypes) was 65535.93, the recessive model was 66537.64 and the dominant model was 66573.9. Hence, we concluded that the recessive model was a better fit than the dominant model.

All analyses therefore support the conclusion that the data is most consistent with a recessive model.

Reviewer #3 (Remarks to the Author):

The authors use pharmacological approach, combined with fluorescence calcium measurement to collect the functional data for CaSR to show urinary/serum biomarkers of calcium metabolism, and performed in vitro studies of cells with reduced DGKD expression. Main purpose is that understanding the genetic susceptibility to kidney stones, which will help to identify potential therapies for kidney stone disease. It is good to see that the Authors have addressed the concerns from the previous version.

Still there are a few minor points that still came to my attention, which I would like to see addressed before publication. These are listed below. Other than that, I think the manuscript has greatly improved and I recommend accepting it for publication.

1. Figure 3B CaSR Blot looks darker background, with 2 bands. Authors need to be marked (}/*) which band they are considering. They should submit a full blot in supplementary. There is no loading control for this western blot, which is necessary.

Thank you for this comment, the CaSR has two bands, one of which is an immature 140kDa form, the other a 160kDa fully glycosylated form. Both bands were considered during densitometry. Initially we used PGK1 as a loading control, we have now changed this to the more conventional α -Tubulin. The full blot is submitted in the 'source-data' file. These changes are noted as follows:

Figure 3 – Main text

Line 466-471 - Main text

“Western blot analyses were undertaken using anti-DGKD (SAB1300472; Sigma), anti-CaSR (5C10, ADD; ab19347; Abcam), and anti-Tubulin (T5168; Sigma). The western blots were visualized using an Immuno-Star Western C kit (Bio-Rad) on a Bio-Rad Chemidoc XRS+ system and relative expression of DGKD and CaSR were quantified by densitometry using ImageJ software. Both the 140kDa immature CaSR band and the 160kDa glycosylated band were considered in densitometry calculations.”

2. In Fluo-4 measurement of Ca²⁺ concentration did they use EGTA in addition to Ionomycin? Authors need to add the details of the data analysis of Ca²⁺ measurement.

Thank you for this comment, cells were stimulated only with Ionomycin, not EGTA. The details of the data analysis are included as follows:

Line 515-526 – Main text

“Changes in intracellular calcium concentrations were recorded via detection of fluorescence for 30 seconds using a PHERAstar microplate reader (BMG Labtech) at 37 °C with an excitation filter of 485 nm and an emission filter of 520 nm. The peak mean fluorescence ratio of the transient response following each individual stimulus was measured using MARS data analysis software (BMG Labtech). Relative fluorescence units were normalized to the fluorescence stimulated by ionomycin to account for differences in cell number and loading efficiency and then further normalised to the maximum response observed for the cells treated with scrambled siRNA^{57,58}. Assays were performed using 4 biological replicates (independently transfected wells, performed on at least 4 different days). Nonlinear regression of concentration-response curves was performed with GraphPad Prism for determinations of maximal response.”

Line 689-693 – Main text

“For in vitro studies statistical comparisons were made with 2-tailed Student’s t-tests, maximal responses were compared using the F-test, and responses at each extracellular calcium concentration were compared using a 2-way ANOVA with Turkey’s multiple-comparisons test using GraphPad Prism 6^{29,58}.”

3. Similarly, in Supplementary Figure 5, “CaSR-mediated responses following DGKD knockdown in HEK-CaSR and HEK-CaSR-NFAT cells” – Details need to add in “Panel E shows intracellular calcium responses of HEKCaSR cells in response to changes in extracellular calcium concentration.”

Thank you for this suggestion, the following text has been added to the figure legend of Supplementary Figure 5:

Line 148- 149 – Supplementary Material

“Responses were normalized to those elicited by stimulation with ionomycin, and subsequently, to the maximum response observed for WT cells.”

4. Concentrations of scrambled (WT) and DGKD (DGKD-KD) siRNA also needs to be added.

Thank you for this suggestion, the following text has been added to the text:

Line 454 – Main text

“Cells were transfected with 10nM scrambled or DGKD siRNA.....”

5. Figure 3D label inside the graph looks odd since nothing indicated in it.

Thank you for this comment. The grey, light red and light blue points shown on Figure 3D represent the post desensitization data. To make this clearer the figure legend has been altered as follows:

Line 768-770 – Main text

“Post desensitization points are shown but were not included in the analysis (grey, light red, and light blue).”

Reviewers' Comments:

Reviewer #2:

Remarks to the Author:

Dear authors,

The conclusion that the association of rs17216707 with kidney stones is most consistent with a recessive model is wrong. The root of this erroneous conclusion is drawn from a counting error in the contingency table for the recessive model in supplementary table 6. The marginal sums for the different genetic models are not shown in supplementary table 6. On close inspection it is clear that the control count in the Recessive model contingency table is 468,508 which is 80,000 more than the total count of 388,508 for the Dominant model which is the correct number (that number is reported in the Methods and in Supplementary Table 7). This error resulted in a greatly inflated effect for the recessive model. Thus the calculation of the recessive model is wrong and the claimed effect of 1.92 is not true.

I suggest that the authors redo supplementary Table 6.

- In supplementary Table 6, what the author refers to as a Full genotypic model is in reality an allelic association test with one degree of freedom which is equivalent to an additive model. Rename the model to "Additive"
- A full genotypic association test with two degrees of freedom should be added to the table and named "Full genotypic model".
- Redo the calculation for the Recessive model. Then a conclusion can then be drawn by comparing the effects of the different models.

In an attempt to reproduce the association of rs17216707 with kidney stones in publicly available UK Biobank data it is clear to me that the association of rs17216707 with kidney stones is most consistent with an additive model. There is a clear effect for both hetero- and homozygotes. However, the effect for homozygotes does not deviate from the additive model (the confidence intervals overlap).

Reviewer #3:

Remarks to the Author:

Authors addressed all the concerns. I would like to congratulate the authors for putting together this nice work. Now the manuscript can be accepted for publication.

1 **Resubmission of: Genetic variants of calcium and vitamin D metabolism in kidney stone disease by**
2 **Howles et al.**

3
4 Reviewer #2 (Remarks to the Author):
5

6 The conclusion that the association of rs17216707 with kidney stones is most consistent with a recessive model
7 is wrong. The root of this erroneous conclusion is drawn from a counting error in the contingency table for the
8 recessive model in supplementary table 6. The marginal sums for the different genetic models are not shown in
9 supplementary table 6. On close inspection it is clear that the control count in the Recessive model contingency
10 table is 468,508 which is 80,000 more than the total count of 388,508 for the Dominant model which is the
11 correct number (that number is reported in the Methods and in Supplementary Table 7). This error resulted in a
12 greatly inflated effect for the recessive model. Thus the calculation of the recessive model is wrong and the
13 claimed effect of 1.92 is not true.
14

15 I suggest that the authors redo supplementary Table 6.
16

17 - In supplementary Table 6, what the author refers to as a Full genotypic model is in reality an allelic association
18 test with one degree of freedom which is equivalent to an additive model. Rename the model to "Additive"
19

20 - A full genotypic association test with two degrees of freedom should be added to the table and named "Full
21 genotypic model".
22

23 - Redo the calculation for the Recessive model. Then a conclusion can then be drawn by comparing the effects
24 of the different models.
25

26 In an attempt to reproduce the association of rs17216707 with kidney stones in publicly available UK Biobank
27 data it is clear to me that the association of rs17216707 with kidney stones is most consistent with an additive
28 model. There is a clear effect for both hetero- and homozygotes. However, the effect for homozygotes does not
29 deviate from the additive model (the confidence intervals overlap).
30

31 We thank the reviewer for noticing this error and apologise for the counting inaccuracy in the contingency table.
32 Supplementary Table 6 has been corrected and amended in line with the reviewer's suggestions, see below:
33
34

35 Supplementary Table 6: Comparison of genotypic models for rs1716707.

Model			N (cases)	N (controls)
Additive	N (allele)	T	10,898	629,730
	N (allele)	C	2,174	147,286
	Odds ratio (95% CI)	1.17 (1.12-1.23)		
	Z-statistic	6.72		
	P-value	1.71×10^{-11}		
			N (cases)	N (controls)
Recessive	N (genotype)	TT	4,551	255,261
	N (genotype)	TC + CC	1,985	133,247
	Odds ratio (95% CI)	1.19 (1.13-1.26)		
	Z-statistic	44.0		
	P-value	3.24×10^{-11}		
			N (cases)	N (controls)
Dominant	N (genotype)	TT + TC	6,347	374,469
	N (genotype)	CC	189	14,039
	Odds ratio (95% CI)	1.26 (1.09-1.46)		
	Z-statistic	3.10		
	P-value	0.0019		
			N (cases)	N (controls)
Full genotypic	N (genotype)	TT	4,551	255,261
	N (genotype)	TC	1,796	119,208
	N (genotype)	CC	189	14,039
	Z-statistic	45.9		
	P-value	1.06×10^{-10}		
			N (cases)	N (controls)
Homozygote	N (genotype)	TT	4,551	255,261
	N (genotype)	CC	189	14,039
	Odds ratio (95% CI)	1.32(1.14-1.53)		
	Z-statistic	14		
	P-value	1.63×10^{-4}		
			N (cases)	N (controls)
Heterozygote	N (genotype)	TT	4,551	255,261
	N (genotype)	TC	1,796	119,208
	Odds ratio (95% CI)	1.18(1.12-1.25)		
	Z-statistic	36		
	P-value	1.95×10^{-9}		

In light of these revised calculations we have altered the text that references Table 6 as follows:

Line 141-144 - Main text

“Analysis of the allelic frequencies of this SNP between cases and controls within the UK Biobank cohort was not able to provide definitive evidence for the recessive model over other genetic models (Supplementary Table 6).”